# Post-surgical adhesions are triggered by calcium-dependent membrane bridges between mesothelial surfaces

Adrian Fischer[1,7], Tim Koopmans[1,7], Pushkar Ramesh[1], Simon Christ [1], Maximilian Strunz[2], Juliane Wannemacher[1], Michaela Aichler[3], Annette Feuchtinger[3], Axel Walch[3], Meshal Ansari [4], Fabian J. Theis [4], Kenji Schorpp[5], Kamyar Hadian [5], Philipp-Alexander Neumann[6], Herbert B. Schiller [2] & Yuval Rinkevich[1✉]

Surgical adhesions are bands of scar tissues that abnormally conjoin organ surfaces. Adhesions are a major cause of post-operative and dialysis-related complications, yet their pathomechanism remains elusive, and prevention agents in clinical trials have thus far failed to achieve efficacy. Here, we uncover the adhesion initiation mechanism by coating beads with human mesothelial cells that normally line organ surfaces, and viewing them under adhesion stimuli. We document expansive membrane protrusions from mesothelia that tether beads with massive accompanying adherence forces. Membrane protrusions precede matrix deposition, and can transmit adhesion stimuli to healthy surfaces. We identify cytoskeletal effectors and calcium signaling as molecular triggers that initiate surgical adhesions. A single, localized dose targeting these early germinal events completely prevented adhesions in a preclinical mouse model, and in human assays. Our findings classifies the adhesion pathology as originating from mesothelial membrane bridges and offer a radically new therapeutic approach to treat adhesions.

[1] Helmholtz Zentrum München, Institute of Lung Biology and Disease, Regenerative Biology and Medicine, Member of the German Center for Lung Research (DZL), Munich, Germany. [2] Helmholtz Zentrum München, Institute of Lung Biology and Disease, Systems Medicine of Chronic Lung Disease, Member of the German Center for Lung Research (DZL), Munich, Germany. [3] Research Unit of Analytical Pathology, German Research Center for Environmental Health, Helmholtz Zentrum München, Munich, Germany. [4] Helmholtz Zentrum München, Institute of Computational Biology, Munich, Germany. [5] Helmholtz Zentrum München, Assay Development and Screening Platform, Institute for Molecular Toxicology and Pharmacology, Munich, Germany. [6] Technical University of Munich, School of Medicine, Klinikum rechts der Isar, Department of Surgery, Munich, Germany. [7]These authors contributed equally: Adrian Fischer, Tim Koopmans. ✉email: yuval.rinkevich@helmholtz-muenchen.de

Pathological adhesions occur when organ surfaces coalesce with and weld to one another or to the walls of their surrounding cavity. Despite improvements in surgical techniques that have reduced trauma; painful and sometimes fatal adhesions still occur in up to 93% of abdominal operations[1–3]. Of patients receiving gynaecological surgery, 64% were readmitted within 10 years for a problem potentially related to adhesions or for further intra-abdominal surgery that could be complicated by adhesions. 2.9% of patients were readmitted for problems directly related to adhesions[4–7]. In emergency surgery that necessitates repetitive operations, severe adhesions can already be seen after the initial operation that either lead to bowel obstruction or completely limit further exploration of the abdomen. Adhesions encountered at surgical re-interventions significantly prolong operative time with time-consuming removal of adhesions and elevated intraoperative and postoperative complications[8,9]. During removal of adhesions, further injuries to adjacent bowel or other organs occur in up to 19% of the procedures, which causes intestinal spillage and severe septic complications[1,8]. For example, after gynecologic surgery, adhesions account for 15–20% of infertility cases[10,11]. This vicious, and sometimes lethal, circle of consecutive operations adds an enormous burden on healthcare, estimated at over $1 billion a year in the United States[11].

In contrast to their clinical impact on patients, pathophysiology of early adhesion formation is still incompletely understood. For example, while the mature stages of the adhesion process, including the activation of immune responses and the formation of macroscopic scars between surfaces are well documented and studied, the mechanisms leading injured organ surfaces to seam remains undisclosed. Furthermore, while adhesions habitually develop from injuries imposed to a single organ surface, the pathology subsequently expands to non-injured and adjacent surfaces, through a mechanism that remains obscure.

Internal organs are covered by an epithelial monolayer called the mesothelium that protects the organs and provides a frictionless interface between them. Although the mesothelium has been proposed to play a role in the adhesion process[12], up until recently the most popular mechanistic model of how adhesions develop was that it was the removal and thus absence of this mesothelium that generates adhesions. It was proposed that without the mesothelium exposed fibroblasts residing in organ interiors can migrate and deposit a fibrin matrix, generating a connective tissue that bonds organ surfaces[13]. However, there was little if any direct evidence for this model, and indeed endometrial and peritoneal surfaces have been shown to adhere rapidly with mesothelium intact[14].

We recently demonstrated, using lineage tracing in mice, that postoperative adhesions form from mesothelial cells and not from fibroblasts depositing matrix[15,16]. However, the cellular and molecular mechanisms remain entirely undisclosed. We hypothesize these events to occur rapidly, preceding scar formation; however these transient early mechanisms are difficult to see in live mice, because the commitment is too quick and events occur too deep beneath the visible surface. Understanding these transient early events, driving rapid organ adherence, and all subsequent pathophysiologic steps, would pave the way toward ameliorating postoperative adhesions in oncology, gynecology, pelvic, and abdominal surgery. Here, we circumvent the inherent difficulties of visualizing germinal events in live animals by developing an in vitro assay that simulates adhesiogenesis between human organ surfaces in microscopic detail, revealing the mechanism of adhesion formation, and transmission from one organ to another. We report adhesions initiate through the rapid generation of membrane bridges that physically tethers cells and organs together. Single-cell RNA sequencing of injured human beads reveals cytoskeletal and calcium-regulating effectors acts as the main components of the early adhesion cascade. Consequently, a single, localized dose targeting these core proteins effectively shuts down adhesions in live mice and our in vitro assay.

## Results

**Early adhesion pathogenesis is recapitulated in vitro.** While the late stages of the adhesion process, including the activation of immune responses and the formation of macroscopic scars between surfaces are well documented, the transient early mechanisms, leading injured organ mesothelial surfaces to weld together, remain completely unknown, as they are too deep and fleeting to see in animals. We thus sought to create an in vitro assay to image the early interactions that occur between organ surfaces and that lead to postoperative adhesions. In brief, we focused on the widely available human mesothelial cell line Met-5A and mixed cells with microcarrier beads (Cytodex® 3—Sigma Aldrich) in a 500:1 ratio to create beads with a mesothelial monolayer surface. After 5 days in culture, beads were isolated through a strainer and subjected to adhesion stimuli (Fig. 1a). Adhesions can be induced by various irritants[17], among which exposure of organs to air during operation, leading to tissue desiccation. Exposing beads to a short bout of ambient air (15 min) in a ventilating cell-culture flow hood led beads to rapidly adhere to a monolayer of cultured Met-5A cells (Fig. 1b). Adhesion onset occurred rapidly, as beads adhered to the monolayer within just 60 min. The mesothelial-covered beads aggregated and continuously recruited more beads over time, until plateauing in adherence capacity at ~72 h. To more accurately assess bead adhesions, and to allow high-throughput screening, we generated cells stably expressing nanoluciferase that were then coated on beads and added to a wild-type monolayer prior to stress (Supplementary Fig. 1A). We then stimulated potential adhesions by administering beads either a desiccation shock, or exposing beads to low concentrations (1 µg/mL) of talcum powder, another clinically relevant adhesion stressor. Both irritants led beads to rapidly adhere to the monolayer, or cluster into large aggregates, reminiscent of the events in adhesions (Fig. 1c, d). Live confocal microscopy of nuclear-labelled cells revealed cells in the monolayers being actively pulled upwards from the culture plate by the carrier aggregates (Fig. 1e and Supplementary Video 1), and scanning electron microscopy of stressed mesothelial aggregates revealed that beads were completely fused, and flattened or buckled (Supplementary Fig. 1e). Importantly, bead adherence in response to tissue desiccation or talcum exposure was specific to their mesothelial coat, and was absent when using HEK-293 human embryonic kidney cells (data not shown). To verify the preclinical applicability of our carrier model, we developed a mouse adhesion model that combines three risk factors known to cause postsurgical adhesions: (1) abrasive damage to organ surfaces due to surgical mishandling; (2) hypoxic pockets that develop at severed vessels and nerves; and (3) talcum powder irritation from surgical gloves[18]. Induction of all three risk factors, together, in wild-type (C57BL/6) mice (see Methods) generated highly reproducible adhesions (Fig. 1f). Early signs of adhesions were visible as early as 24 h post injury, manifesting as fragile attachments between organ surfaces. After 3–5 days, adhesions had spread to adjacent surfaces of the abdominal wall, often attached to secondary uninjured organs, such as abdominal fat or liver (Fig. 1f). Importantly, in vitro carrier aggregates and in vivo murine adhesions, both, expressed the same characteristic adhesion proteins (mesothelin (MSLN), CD44, and hypoxia-inducible factor 1-alpha) (Supplementary Figs 1C and 2A). Adhesion development in vitro followed the same sequence of events as seen in animals. We observed in vitro

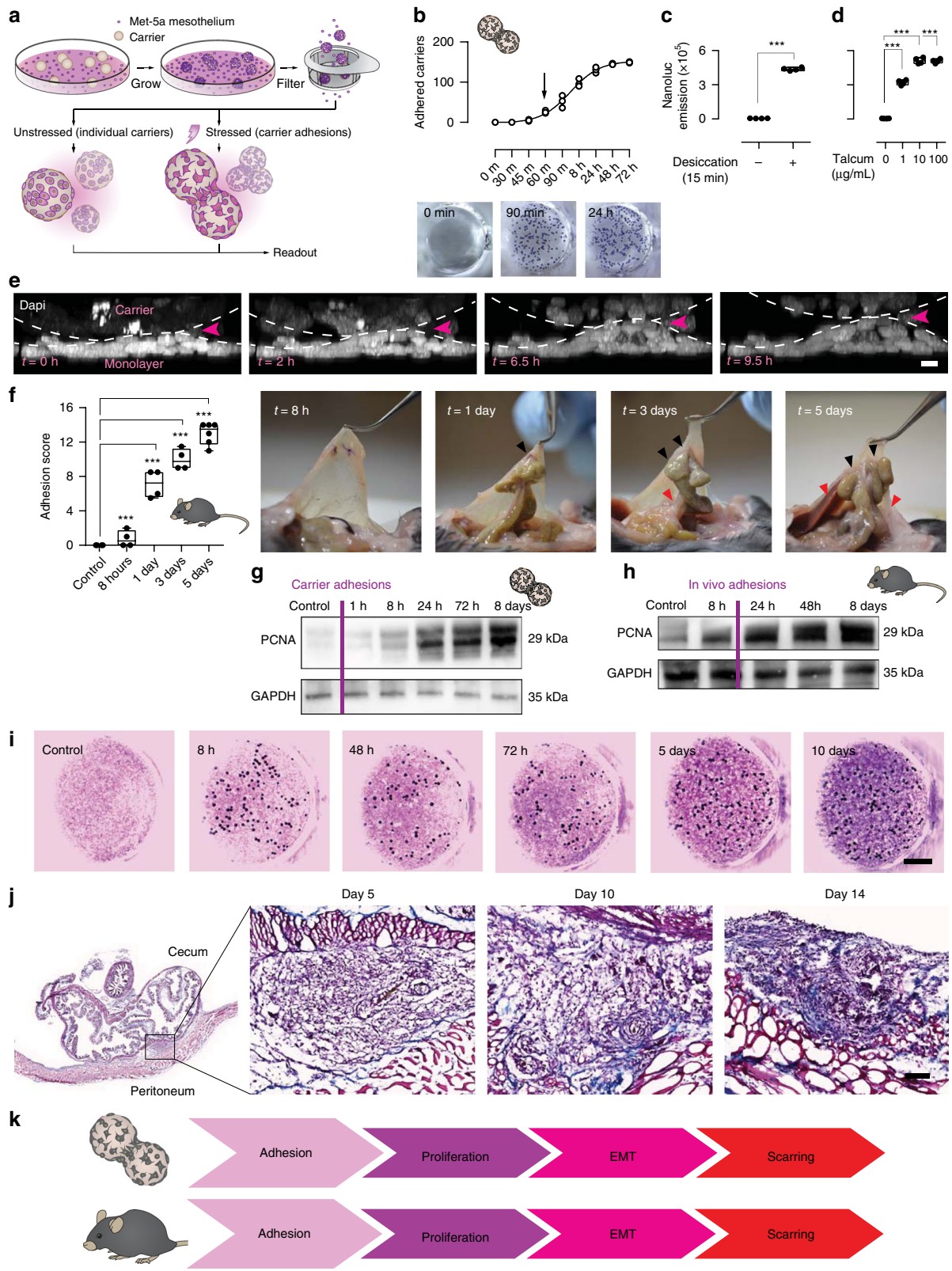

adhesions initiating with rapid adherence between beads, followed by mesothelial proliferation (Fig. 1g). The same steady increase in mesothelial proliferation occurred in vivo, ultimately developing into a multilayered and thickened surface (Fig. 1h and Supplementary Fig. 2A, B). Markers for epithelial-to-

mesenchymal transition (EMT) including Wilms' tumor 1 (WT1), and the mesenchymal marker alpha-smooth-muscle actin (α-SMA) emerged after proliferation onset in carrier aggregates (Supplementary Fig. 3A). In vivo, light-sheet and confocal images confirmed that the same EMT markers WT1, α-SMA, Slug/Snail,

**Fig. 1 Microcarrier model recapitulates physiological adhesions. a** Overview of the bead assay. **b** 15 min desiccation shock induces carrier-to-monolayer aggregation, which develops as fast as 60 min after injury. Three biological replicates. **c, d** Nanoluciferase assay to measure adhesion propensity. Desiccation shock and talcum powder both induce carrier-to-monolayer aggregation. Four biological replicates; ***$p < 0.001$, two-tailed Mann–Whitney (**c**) and Kruskal–Wallis followed by Dunn's (**d**). **e** Side view of live imaged bead-monolayer adhesion showing exerted pulling forces. Scale bar, 10 μm. Representative images of three biological replicates. **f** Adhesion severity (see Methods) increases with time. Black arrows, suture sites. Red arrows, secondary organ attachments. Four biological replicates; ***$p < 0.001$, two-tailed Mann–Whitney. **g** Immunoblot of lysed Met-5A cells, various time points after a 15 min desiccation shock. GAPDH serves as loading control. Representative images of three biological replicates. **h** Immunoblot of excised murine adhesion tissue, various time points after injury. Representative images of three biological replicates. **i** Delayed matrix deposition in vitro in stressed carriers (dark spheres)-to-monolayer (Masson Trichrome staining). Representative images of ten biological replicates. Scale bar, 500 μm. **j** Masson Trichrome of adhesion tissue section 5–14 days after injury. Representative images of three biological replicates. Scale bar, 100 μm. **k** Schematic overview of the sequence of events characteristic of adhesion development in both the carrier assay and in vivo model. Error bars represent standard error of the mean.

CD44, and MSLN were upregulated at injury sites where adhesions had developed (Supplementary Fig. 3B, C), and emerged subsequent to the initial adhesion and proliferation events (Supplementary Fig. 3D). Similarly, rapid bridging between organs did not require any matrix deposition. Fibrillar protein synthesis, such as collagens and fibronectin developed well after adhesions had formed, peaking around 5–10 days post surgery (Fig. 1i, j, and Supplementary Fig. 3A). The above data firmly establish that our in vitro model recapitulates the adhesion process, and that the bridging of organ surfaces represents an independent early event preceding proliferation, EMT, and matrix deposition (Fig. 1k).

**Profuse membrane bridges between mesothelial surfaces initiate adhesions**. Our newly established in vitro adhesion model thus allowed us to view the process of adhesion formation in real-time. Strikingly, when mesothelial cells were grown in Matrigel after a 15 min desiccation shock, a vast network of protrusions between cells emerged that bridged nearby colonies, whereas healthy cells grew cystically as separate colonies (Fig. 2a). To view the interface between mesothelial surfaces at higher resolution we performed live imaging via 3D interference reflection confocal microscopy. We observed stressed mesothelia continuously reaching out toward one another through highly dynamic protrusions during the first step of adhesion development (Fig. 2b and Supplementary Video 2). To further zero in on the protrusions, we transfected mesothelial cells with Lifeact-eGFP or -mCherry, coated them on beads, and performed multiphoton live imaging. Sites of interface between stressed beads were interconnected by assemblages of membrane protrusions that radiated from cells in three-dimensions (Fig. 2c). Similarly, confocal videoing of stressed cells expressing stable membrane-bound eGFP or dTomato revealed a highly dynamic cytoskeleton with various forms of membrane protrusions that continuously engaged their surroundings (Supplementary Video 3). Using machine learning algorithms (see Methods), we quantified protrusions in stressed and unstressed cells and documented a 63% increase in protrusion coverage in stressed mesothelium (Supplementary Fig. 4B), with a range of widths and lengths (Supplementary Fig. 4C). Stressed mesothelia exhibited a wide range of protrusion types (Supplementary Fig. 4D), including an additional unreported membrane protrusion type that occupied up to 25% volume of the main cell body and that was ladened with secondary protrusions, all moving independently from the main cell body. We termed these 'akropodia', from the Greek word 'ákros', referring to extremities such as hands (Supplementary Fig. 4D–F). These coalitions of membrane protrusions enabled extensive membrane contacts to develop between surfaces (Supplementary Video 4). As a result, adherence force measurements (see 'Methods') showed stressed cells displayed more than threefold increase in binding strength compared to unstressed

cells (Supplementary Fig. 4G, H), equivalent to a considerable force of 1.14 g/cm². These analyses indicate that even a small surface injury is sufficient to twine large organ surface areas. Scanning electron microscopy images of mouse peritoneal adhesion tissue revealed the same morphologic transformations at sites of injury as occurred in our in vitro human assays. Stressed cells were clearly visible in vivo at sites of injury as early as 16 h after injury, when adhesions had not yet developed, with extensive membrane protrusions, contrary to uninjured sites that retained a typical cobblestone morphology (Fig. 2d). Similarly, multiphoton microscopy imaging revealed profuse PDPN+ cells at sites of injury, with membrane protrusions that probed in three-dimensional space (Fig. 2e).

To irrefutably prove that protruding cells and postsurgical adhesions originate directly from mesothelial cells in vivo, and not from existing fibroblasts, we traced the fate of injured mesothelium using Procr[CreERT2-IRES-tdTomato] (hereafter referred to as PROCR) knock-in mice, in which a CreERT2-IRES-tdTomato cassette is inserted after the first ATG codon of Procr[19]. We have found Procr to be highly specific to mesothelial surfaces without labelling resident fibroblasts (unpublished). We then crossed these mice with a reporter Rosa26[mTmG] line to mark all Procr descendants as GFP positive (Supplementary Fig. 5A). Three consecutive tamoxifen injections (2 mg per injection) administered intraperitoneally (Supplementary Fig. 5B) resulted in ~50% label in mesothelial surfaces (Supplementary Fig. 5C). Tissue sections of the peritoneal and cecal wall showed specific labelling in the surface mesothelium after tamoxifen administration, co-expressing PDPN (Supplementary Fig. 5D). To capture the adhesion phenotype in real-time in vivo, we performed live imaging of injured peritoneal mesothelium 4 h after injury, when adhesions have not yet formed. We observed loss of cell–cell junctions and protrusion development 4–12 h after adhesion induction in vivo (Fig. 2f and Supplementary Video 5). Consequently, 5 days after adhesion induction, GFP+ mesothelial cells were seen between the fused parietal and visceral layers, with a host of protrusions emanating in a radial pattern (Fig. 2g and Supplementary Video 6). Adhesion tissue from these mice co-stained for the pan-immune and pan-fibroblast markers CD45 and PDGFRα, respectively, showed some GFP+ cells also expressed PDGFRα (while being negative in healthy tissues), indicating that at 5 days post injury, injured mesothelium has adopted fibroblast properties (Supplementary Fig. 5E). As expected, no GFP+ cell expressed CD45, further supporting a mesothelial rather than immunological cellular origin for adhesions (Supplementary Fig. 5F). Finally, we crossed PROCR mice with Rosa26[tm1(DTA)Lky] mice to selectively ablate PROCR+ mesothelial cells upon tamoxifen administration (Fig. 2h). Adhesions completely failed to develop in mice treated with a single dose of tamoxifen immediately after surgery, whereas genotype-negative animals generated full blown adhesions

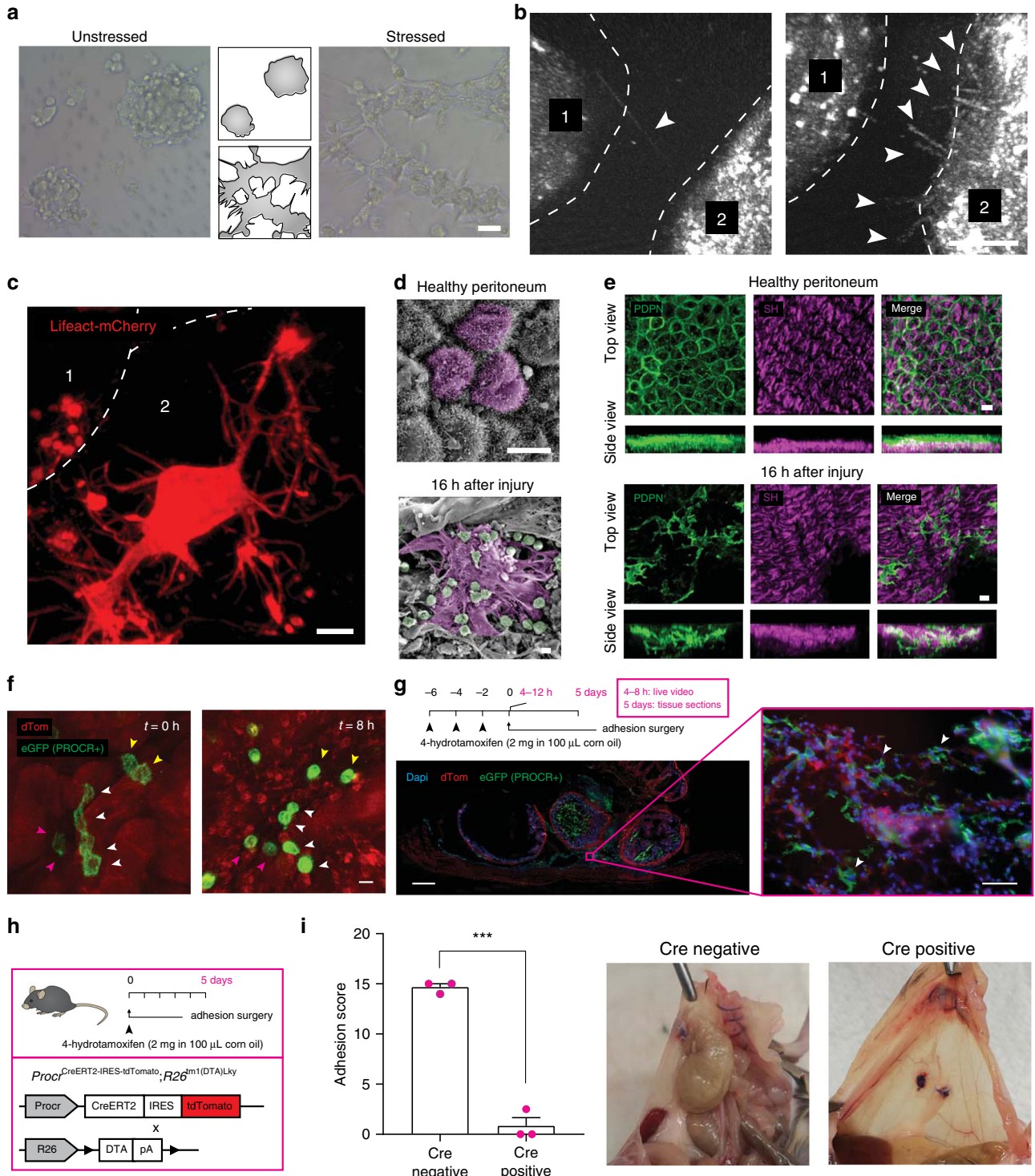

**Fig. 2 Mesothelia produce cytoskeletal protrusions to bind and transmit adhesive potential. a** Phase-contrast image and corresponding silhouette of desiccation-stressed and unstressed Met-5A cells seeded on a matrigel bedding. Representative images of four biological replicates. Scale bar, 30 μm. **b** Whole-mount 3D reflectance confocal live imaging of two stressed mesothelial cells grown on beads, showing connecting nanotubes. Representative images of three biological replicates; Scale bar, 3 μm. **c** Transient labeling of actin filaments through Lifeact-mCherry showing protrusion networks at carrier–carrier contacts. Representative images of three biological replicates; Scale bar, 10 μm. **d** Scanning electron microscopy image of healthy and injured mouse peritoneum. Color overlay based on morphology. Representative images of four biological replicates; Scale bar, 5 μm. **e** Multiphoton image of whole-mount top and side view of healthy and injured PDPN+ peritoneum. SH second harmonics, showing collagen bed. Representative images of three biological replicates; Scale bar, 15 μm. **f** Multiphoton live video of tamoxifen-treated Procr^CreERT2-IRES-tdTomato;Rosa26^mTmG mouse peritoneum, showing individual mesothelial cells. Representative images of three biological replicates; Scale bar, 10 μm. **g** Confocal image of tamoxifen-treated Procr^CreERT2-IRES-tdTomato; Rosa26^mTmG mouse adhesion tissue section, 5 days after injury. Representative images of three biological replicates; Scale bar, 1000 μm (left) and 50 μm (right). **h** Overview of the tamoxifen regime and the Procr-DTA transgene. **i** Adhesion score of tamoxifen-treated Procr-DTA mice 5 days after injury. Three biological replicates; ***p < 0.001, two-tailed Mann–Whitney.

(Fig. 2i). Collectively, these results prove that mesothelial cells are precursors of protruding cells and of adhesions in vivo, with minimal contributions from fibroblast cell populations.

**Adhesion pathogenesis is transmitted through protrusion-based membrane fusions.** Surprisingly, stressed mesothelial cells in our in vitro adhesion model were frequently observed to express both dTomato/mCherry and GFP, indicating that membrane contacts were followed by fusions and exchange of cytoplasmic content during early stages of adhesions (Fig. 3a, Supplementary Video 6). Strikingly, when unstressed mesothelial

beads were exposed to stressed cells, and then separated, they were fully able to generate adhesions even though they had never experienced stress (Fig. 3b, see Methods for more details). Transmission of the adhesion phenotype could occur a total of three rounds after the original stress stimulus (Fig. 3b), indicating cytoplasmic exchange between cells is coupled with active signal transduction. We never observed cells with multiple nuclei (data not shown), suggesting fusions may be limited to the protrusion compartment. As proof, we generated two additional cell lines, one expressing Cre recombinase and one expressing a stop codon flanked by two LoxP sites followed by a dTomato-

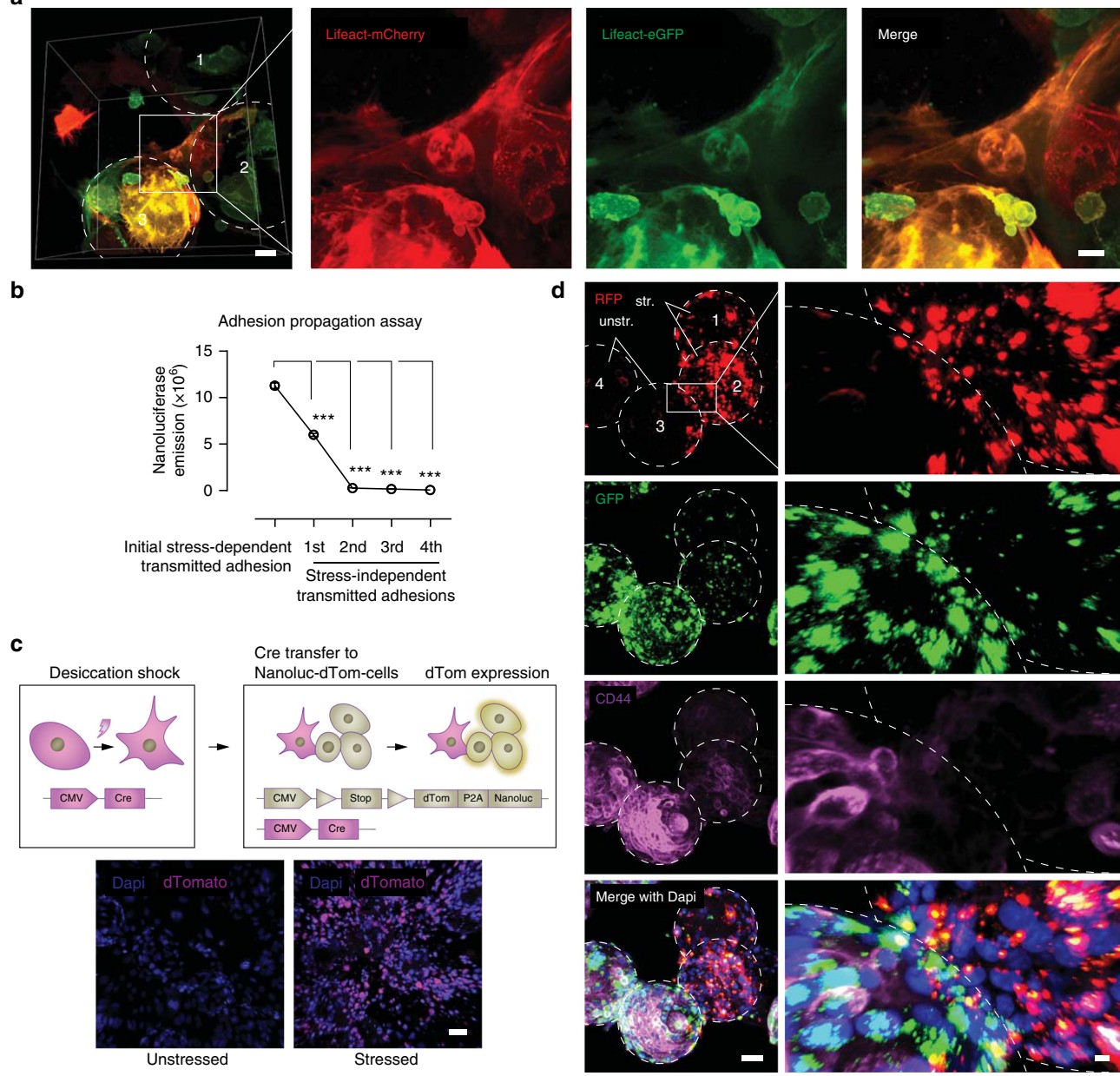

**Fig. 3 Protrusions are capable of membrane fusion and transmission of cytosolic contents. a** Whole-mount multiphoton image of bead clusters seeded with Lifeact-mCherry and -eGFP transfected cells, 24 h after desiccation, showing cells with double GFP and mCherry expression. Representative images of three biological replicates; Scale bar, 10 μm. **b** Adhesion propagation assay (see Methods) with nanoluciferase expressing Met-5A cells, showing transmittance of the adhesion phenotype 24 h after desiccation, and thereafter every 3 h. Three biological replicates; ***$p < 0.001$, One-way ANOVA followed by Tukey. **c** Carrier-monolayer confocal image showing exchange of Cre-protein driving dTomato expression in receiving cells after 24 h. Representative images of ten biological replicates; Scale bar, 50 μm. **d** Whole-mount antibody stain of CD44 (magenta), and lipid membrane dyes Dio (green) and PKH26 (red) 24 h after stress. Red-colored beads were originally stressed by desiccation, whereas green-colored beads never experienced stress. Representative images of three biological replicates; Scale bar, 50 μm (left) and 10 μm (right). Error bars represent standard error of the mean.

P2A-Nanoluciferase. In this transmission assay, Cre recombinase only drives dTomato-P2A-Nanoluc expression if extensive cytoplasmic mixing occurs that allows Cre recombinase protein to be transferred and shuttled into the nucleus of the dTomato-P2A-Nanoluc cells. We incubated stressed mesothelial beads onto an unstressed mesothelial monolayer for 3 h and observed extensive dTomato fluorescence 24 h later, indicating extensive cytoplasmic mixing had indeed occurred at this point (Fig. 3c). To expand on these findings, we labeled stress-exposed carrier-bound cells with red membrane lipid dyes, labeled unstressed carrier-bound cells with green membrane lipid dyes, and then proceeded to mix both cell populations. After 24 h, we observed two major events: (1) originally unstressed (green) cells were frequently seen migrating on stressed carriers, co-expressing red label, indicating active membrane mixing with an adhesion phenotype; and (2) the originally unstressed carrier-bound cells formed tethering scaffolds themselves, and immuno-labelling showed these cells expressed CD44, a surface protein exclusively expressed on stressed cells, confirming transmissibility of the adhesion phenotype (Fig. 3d). Taken together, these data indicate that adhesions are formed by fusion-capable membrane protrusions between mesothelial surfaces, and that the adhesion's underlying pathologic cell phenotype is transmittable upon contact with healthy mesothelial surfaces.

**Pathway analysis identifies the key steps in adhesion initiation**. To determine the transcriptional program that imparts adhesion capacity to healthy mesothelial cells, we analyzed subtle gene-expression changes by performing highly parallel single-cell RNAseq genome-wide expression profiling of individual mesothelial cells exposed to stress (Supplementary Fig. 6A) using the Drop-seq workflow[20]. We individually sequenced >16,000 cells from Met-5A mesothelial cells at various time points after exposure to a 15 min desiccation shock, as well as under control unstressed conditions. Using unique molecular identifier barcode counting[21], we quantified 20,027 genes and performed principal component analysis with the count levels of 7942 genes with the biggest difference between the groups (see Methods). Stressed cells clustered separately from unstressed cells after 8 h and onwards, indicative of an abrupt phenotypic change (Fig. 4a and Supplementary Fig. 6B). Most cells from later time points clustered together, suggesting that major transcriptional changes occurred during the first 8 h post stress (Fig. 4a).

Additional computational methods, including partition based graph abstraction and differential gene-expression analysis using a likelihood-ratio test for single cell gene-expression data[22] (see Methods) independently corroborated these findings (Fig. 4b and Supplementary Fig. 6C). Close inspection of the top upregulated genes in the 8 h cluster showed a clear biphasic response, with strong expression at 8 h, and virtual absence thereafter, further highlighting the first 8 h after stress as the decisive time point at which adhesiogenesis is initiated. Insight into the function of these genes allowed us to partition them into two distinct functions: (1) actin cross-linkers and cytoskeletal modulators, and (2) calcium regulators (Fig. 4c, d), which we corroborated by gene pathway analyses (Fig. 4e, f).

Importantly, many of the protein products of the differentially expressed genes, all of which interact and modulate the actin cytoskeletal network or calcium homeostasis, were highly upregulated in stressed cells, and virtually absent from unstressed cells, including MYL9 (myosin light chain 9, a calcium-sensitive regulatory protein that is necessary for cytoskeletal dynamics[23]), Cofilin (an actin disassembly protein), Ezrin (an A-kinase anchoring protein that links the membrane with the actin cytoskeleton), ARF-GAP1 (ADP-ribosylation factor GTPase-

activating protein 1, a Golgi-associated enzyme that regulates actin architecture[24]), Rho GTPases (a family of well-known G proteins that control intracellular actin dynamics and cytoskeletal programming[25]), and AKAP12 (A-kinase anchor protein 12, a compartmentalizing protein that localizes at the membrane and is regulated by intracellular calcium[26] (Supplementary Fig. 7A). Similarly, adhesion-prone cells were identifiable in vivo by high expression of the same battery of 4 markers: MYL9, ARF-GAP1, Rho GTPases, and AKAP12 (Supplementary Fig. 8A). Just like our human in vitro adhesion assay, these four markers were upregulated under adhesion conditions, and were completely absent in naive mesothelium. To directly test if calcium signaling is altered during adhesiogenesis, we introduced constructs in vitro either expressing the calcium reporter GCaMP6s or permeable calcium reporter X-Rhod-1 into our in vitro assay and both revealed elevated amounts of accessible calcium upon stress (Supplementary Fig. 7B, C). To verify the clinical relevance of our marker findings, we analyzed histologic sections of abdominal adhesions from patients undergoing surgery. All human adhesion tissues tested stained positive for our battery of adhesive cell markers CD44, ARF-GAP1, pan Rho, AKAP12, and MYL9 (Fig. 4g), indicating that these adhesion cell phenotypes are conserved across mouse and human adhesion tissue.

**Blocking membrane protrusions inhibits postoperative adhesions**. To identify adhesion preventing agents we pursued two independent approaches. The first—an unbiased in vitro screening approach using our bead assay combined with a curated library of 1280 FDA-approved small molecules (Fig. 5a). The second—an in vivo targeted approach where we analyzed the functional involvement of 31 targets from our RNAseq dataset using small-molecule inhibitors and blocking antibodies. Amongst the small-molecule hit candidates, our FDA screen highlighted the calcium channel blocker (CCB) Bepridil as a potent inhibitor of surgical adhesions (Fig. 5a). Several additional non-FDA-approved small molecules, Rhosin (inhibits the Rho-GEF binding domain), CK-666 (inhibitor of the Arp2/3 complex and actin assembly), and Golgicide A (inhibits Arf1-mediated actin organization) were similarly capable of completely preventing adhesions in our bead assay (Fig. 5b). Importantly, all four compounds blocked pathologic cell transmissions to healthy surfaces (Fig. 5c) and the ability to generate protrusions (Fig. 5d). To confirm the results of our pharmacology screen, we treated stressed and unstressed cells with a panel of RNA probes targeting core adhesion genes we identified in our RNAseq dataset. Targeted mRNA degradation through these probes effectively prevented bead adherence (Fig. 5e and Supplementary Fig. 9A).

To generalize our mechanistic findings, we replaced the Met-5A mesothelial cell line with primary mesothelial cells that were isolated from the abdomen using magnetic bead cell sorting (see Methods), and cultured them in beads soaked in fluorescent dyes. Similar to Met-5A cells, PDPN+ primary mesothelial cells readily covered beads (Supplementary Fig. 9B), and adopted the stressed phenotype upon a 15 min desiccation shock, with protrusions extending in a radial pattern (Supplementary Fig. 9C). Critically, primary abdominal mesothelial cells developed adhesions between beads in a time frame matching that of Met-5A cells (Supplementary Fig. 9D, E), and were unable to develop adhesions in the presence of our panel of pharmacological compounds (Supplementary Fig. 9F). These findings indicate that calcium-dependent membrane protrusions are universal germinal events driving adhesions, irrespective of organ type.

Next, we confirmed the specificity of our findings in vivo by comparing the effects of our panel of small-molecule inhibitors (Rhosin, CK-666, Golgicide A, Bepridil), to a battery of 27

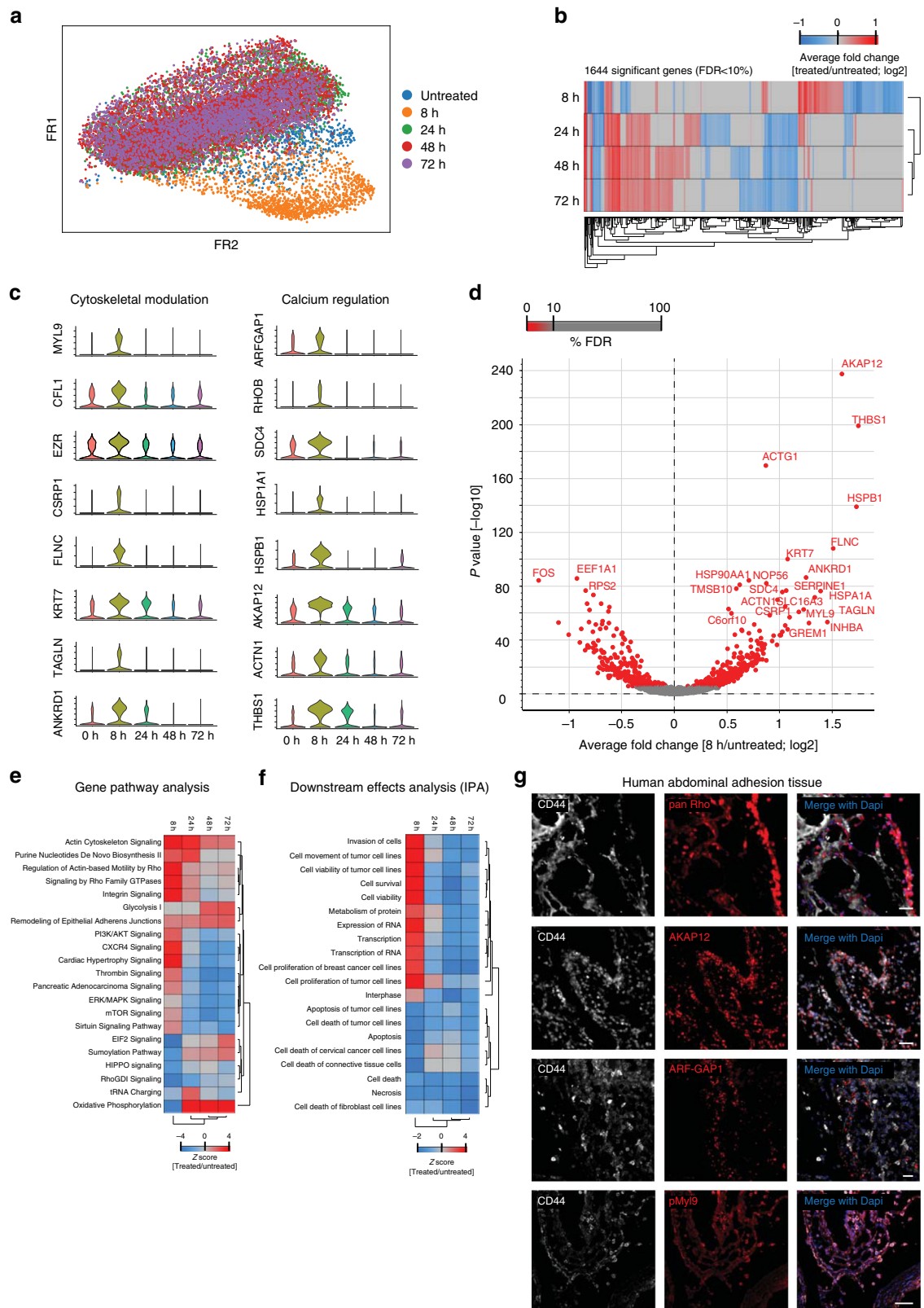

**Fig. 4 Single-cell RNAseq identifies cytoskeletal effectors as core mesothelial program. a** t-SNE visualization of >16.000 Met-5A cells colored by group (time after injury) assignment. **b** Principal component analysis of the top 1644 significantly expressed genes showing separation of the 8 h experimental group. **c** Violin plots showing normalized expression (scored by UMIs) of cytoskeletal and calcium genes peaking 8 h after desiccation shock. **d** Volcano plot of top regulated genes 8 h after desiccation shock. **e**, **f** Gene pathway and downstream effects analysis (IPA). **g** Confocal image of human CD44+ abdominal adhesion tissue sections immuno-stained for core adhesion markers. Scale bar, 100 μm.

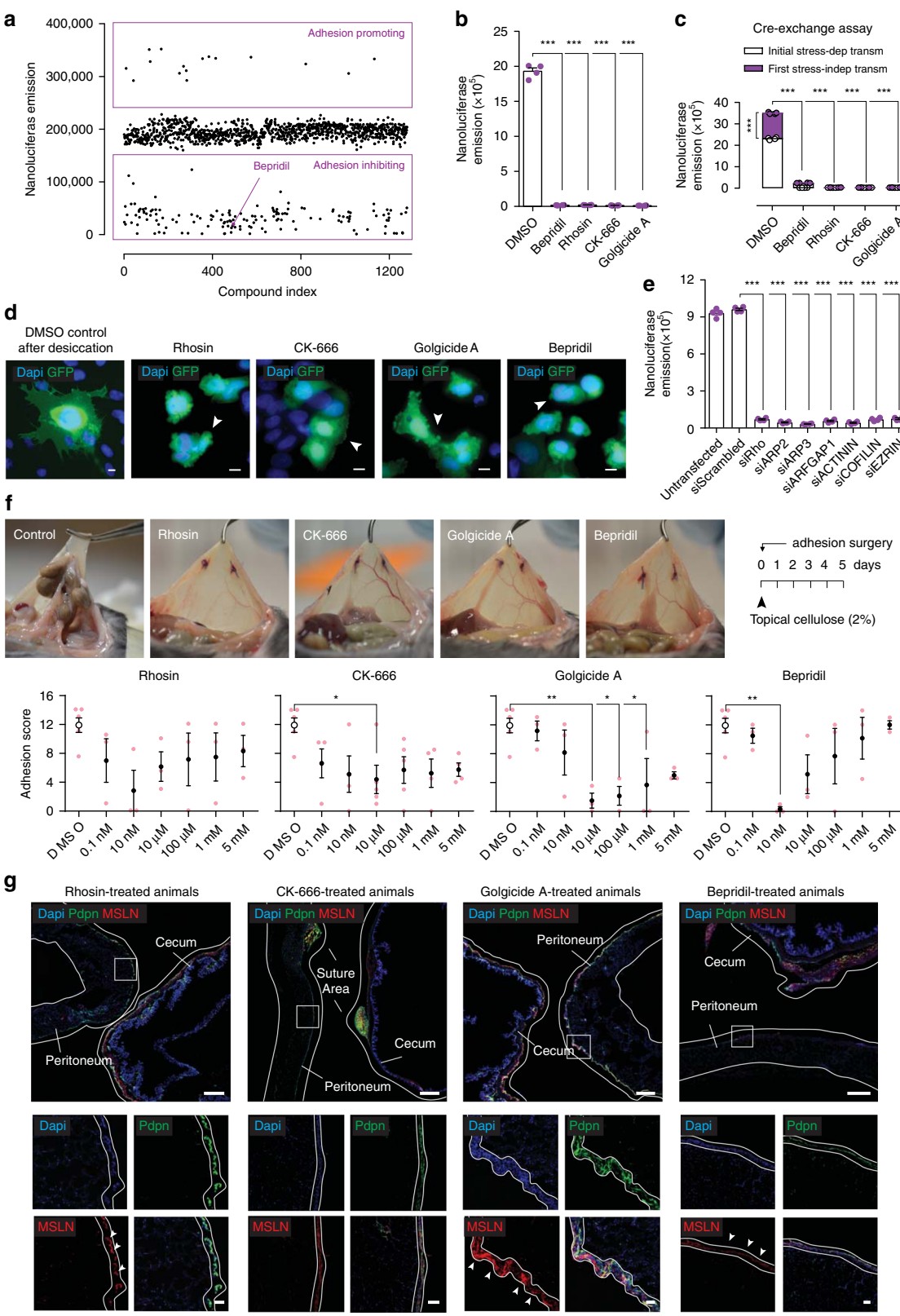

additional transcriptional and signaling pathway targets. These 27 targets included WNT, Notch, and ERK signaling, which we identified as being upregulated in our single-cell transcriptomics analyses, but whose peaks did not confine to the 8 h time point. Following adhesion induction, mice received compounds through daily intraperitoneal injection, and were sacrificed at day 5 for analysis. The actin modulators CK-666, Rhosin, and Golgicide A,

and the calcium channel antagonist Bepridil all robustly inhibited adhesion formation, whereas inhibition of a broad range of signaling targets through the other 27 compounds did not (Supplementary Fig. 10A). These results confirm our in vitro findings that calcium-dependent cellular protrusions drive adhesion pathogenesis. Finally, we explored optimal administrative routes for our adhesion prevention agents using a single

**Fig. 5 Blocking protrusions prevents adhesion development. a** In vitro adhesion assay screening using the Prestwick library consisting of 1280 FDA compounds. **b** Nanoluciferase adhesion carriers-to-monolayer assay 24 h after desiccation, and after treatment with small molecules targeted against core adhesion genes (10 μM). Four biological replicates; ***$p < 0.001$, two-tailed Mann–Whitney. **c** Cre-exchange assay (see Methods) with nanoluciferase expressing Met-5A cells after treatment with small-molecule inhibitors for 24 h (10 μM). Four biological replicates; ***$p < 0.001$, two-tailed Mann–Whitney. **d** Epi-fluorescent representative images of Met-5A cells stably expressing membranous GFP, stressed with desiccation and treated with small-molecule compounds for 24 h (10 μM), showing impaired protrusion development. Representative images of ten biological replicates; Scale bar, 10 μm. **e** Nanoluciferase adhesion carriers-to-monolayer assay 24 h after desiccation, and after treatment with small-interfering RNA against core adhesion genes (1 μg). Four biological replicates; ***$p < 0.001$, two-tailed $t$ test. **f** Adhesion score 5 days after injury, of mice treated with small-molecule compounds dissolved in 2% cellulose that was applied topically at the injury site once before closure. Four biological replicates; *$p < 0.05$, **$p < 0.01$, One-way ANOVA followed by Tukey. **g** Adhesion tissue sections derived from (**f**), immuno-stained for EMT and mesenchymal markers. Representative images of four biological replicates; Scale bar, 500 μm (overview) and 50 μm (inlet). Error bars represent standard error of the mean.

topical administration of our target molecule after surgery in mice. We topically applied each of the above four compounds through a viscous solution of 2% cellulose that was placed immediately after injury onto the injured area alone. Remarkably, a single early and localized administration of a 10 nM gel formulation effectively and completely inhibited adhesion formation in animals (Fig. 5f). Consistent with our in vitro findings of cell protrusions acting early and independent from all other pathologic events, we found EMT (marked by WT1, MSLN, and αSMA expression), and mesothelial surface thickening, to be present on injured treated organ surfaces (Fig. 5g and Supplementary Fig. 11A, B), even though adhesions failed to develop in these mice.

Our findings reveal that membrane protrusions and fusions between mesothelial surfaces are the early germinal events driving surgical adhesion formation, and provide a new therapeutic window and preventative approach to curtail adhesion formation across a range of surgical settings (Fig. 6a).

## Discussion

Various anti-adhesion devices, agents, and strategies have been developed over recent years[27,28]. However—despite the fact that adhesion formation still represents one of the main burdens of surgical care—none of these recent developments has entered clinical routine use. One primary reason is their clinical inapplicability, but also the fact that adhesion formation is incompletely understood. Here we discover that adhesions are caused by membrane protrusions and fusions between mesothelial surfaces, both of which rely on calcium signaling. These early events precede the currently known stages of mesothelial proliferation, thickening, and scar formation.

Although the presence of CD45+ immune and PDGFRα+ fibroblastic cells in adhesion cores indicate their involvement in the adhesion process, we believe these populations serve mainly to modulate and likely exacerbate the adhesion response, but are not strictly required for the initiation of it. Furthermore, in our PROCR+ transgenes of end stage adhesions, GFP-labelled cells express PDGFRα (whilst being PDGFRα negative under healthy control conditions), indicating that they have adopted fibroblast properties. Thus, while the fibroblast is unmistakably an endpoint player in the adhesion pathology, we believe that the bulk, if not all of these cells have a mesothelial origin. On top of that, none of the GFP-labelled mesothelial cells show marker expression normally attributed to immune lineages, indicated by their lack of CD45 expression, suggesting adhesiogenic cells have no immunological origin. Although, we did not perform a comprehensive study to definitively rule out the contributing factors of these lineages, collectively our in vitro assays, in vivo lineage tracing, live imaging of injured mesothelium, and selective ablation of PROCR+ mesothelium, confirms the mesothelial adhesion program as the critical and major player in generating abdominal

adhesions, as well as provide a conceptual framework as to how these adhesions develop.

Targeting this newly discovered early adhesion program is ideally suited to prevent adhesions in patients that have undergone surgeries or are about to receive dialysis. Among the many exciting new therapeutic avenues this opens up, our blocking experiments reveal inhibitors of calcium signalling as potential drugs to overcome adhesion pathogenesis. Specifically, Bepridil hydrochloride is an FDA-approved drug that effectively inhibits adhesion formation at nanomolar ranges. Bepridil hydrochloride is an antianginal drug classified as a CCB[29,30]. It is described as a long-acting nonselective CCB targeting both L- and T-type calcium channels[31]. Chemically it is not related to other CCBs such as diltiazem, nifedipine, and verapamil. Bepridil was approved in the early 90 s by the FDA for treating angina pectoris and was marketed by Johnson and Johnson in the US. Although Bepridil effectively reduced angina, patients who received long-term (4–12 weeks) treatment with Bepridil (200–600 mg, once a day, orally)[32] showed signs of inducing QT interval prolongation and torsades de pointes (TdP)[33]. Due to its cardiovascular side effects, specifically TdP, Bepridil was discontinued in the United States. However, Bepridil is still widely used in other countries[34]. Despite the side effects induced by daily, systemic, and high concentrations of Bepridil, we believe that short-term use may result in fewer adverse effects and may be less problematic than that observed in angina patients requiring long-term treatment (>4 weeks). Most of the patients that developed TdP had received relatively high doses of Bepridil[35], whereas more recent clinical reviews on patients with atrial fibrillation that received Bepridil treatment, propose not to exceed a dose of maximum 200 mg/day[34]. The risk-to-benefit value to the patient of short-term use of Bepridil would of course require careful consideration and would have to be implemented with caution to account for the potential side effects. Alternatively other more specific CCBs could be tested whether they show comparable or even better anti-adhesion effects. Since our animal experiments combine three risk factors to induce adhesions, our findings likely uncover a universal mesothelial response to injury. The fact that adhesions culminate in response to diverse injuries and are ubiquitously manifested in all body cavities and all organ surfaces further implies for the general applicability of these four compounds as pan-inhibitors for adhesion prevention.

## Methods

**Cell culture**. Met-5A cells were cultivated in 10% FBS (Sigma Aldrich, #F9665), F199 (Sigma Aldrich, #M4530), 18 ng/mL EGF (R&D systems, #236-EG), 400 nM Hydrocortisone (Sigma Aldrich, #H4001), 16 ng/mL insulin-transferrin-selenium (Gibco, #41400045), 10 mM HEPES (Gibco, #15630080), 2.5 mg/L amphotericin (Gibco, #5000980), trace elements B (Corning, #15343641), 50 units Penstrep (Gibco, #15070063). After desiccation shock, cells were cultured in 'assay medium': 2% FBS, 10 mM HEPES, trace elements B and 50 units Penstrep. Cells were cultivated on 2% gelatin (Sigma Aldrich, #G1393) coated dishes. Cells were passaged using PBS and Trypsin-EDTA (Sigma Aldrich, #T4049).

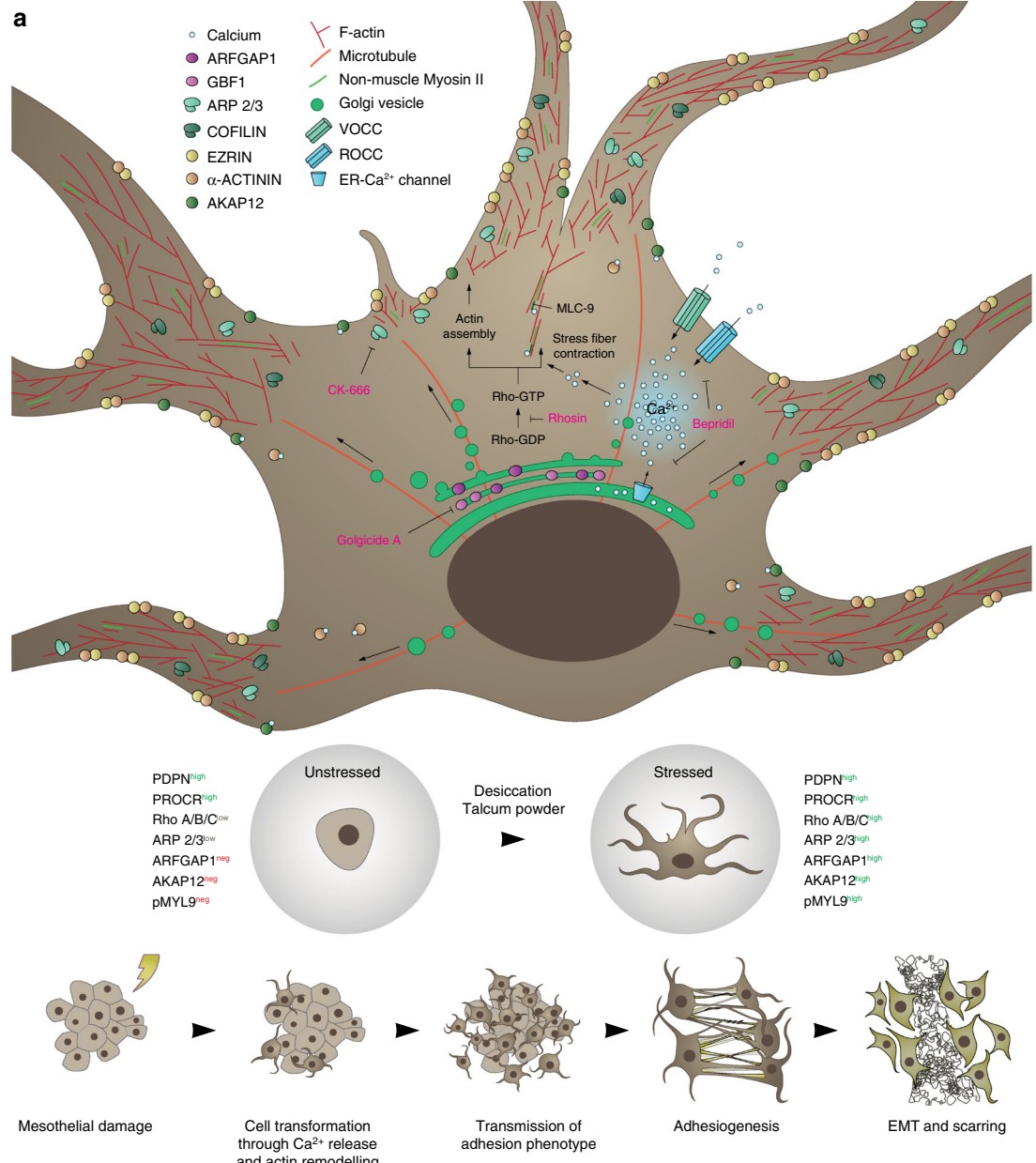

**Fig. 6 Proposed model for the early events driving adhesiogenesis. a** Injury to a serosal layer induces a dramatic and rapid shift in mesothelial morphology through the formation of cytoskeletal protrusions. These allow for (1) the physical binding and fusion to neighbouring healthy cells (e.g., at apposing serosal surfaces), and (2) transmission of pathological behavior. This initiates organ tethering and rapid spread of adhesions through serosal surfaces. Once established, mesothelia commit EMT and deposit matrix to form a macroscopic scar.

**Stable cell lines**. Cells were transfected with corresponding PiggyBac- and Helper plasmids using Lipofectamine 2000 (Invitrogen, #11668) according to the manufacturer's instructions. After 48 h the medium was replaced with 10 ng/mL Puromycin (Tebu-Bio, #BIA-P1230) containing medium. Every other day medium was replaced, up until 2 weeks of selection, after which transgenes were stably expressed.

**Primary cell isolation and culture**. 5 mL of pre warmed dissociation medium (Dulbecco's modified Eagle's medium (DMEM, Life Technologies, #10565-042), Collagenase IV (50 mg/mL, Worthington Biochemical), and 20 $\mu$M CaCl$_2$ was injected into the abdomen of freshly sacrificed wild-type C57BL/6JIV mice. After 5 min the resulting cell suspension was collected, strained through a 100 $\mu$m filter, and spun and washed with 2% fetal bovine serum (FBS, Sigma Aldrich, #F9665) in PBS. Cells were then sorted using MACS® Technology (Miltenyi Biotec). PDPN+ LYVE1− CD31− CD45− cells were cultivated in DMEM supplemented with 10% FBS and 50 units Penstrep (Gibco, #15070063).

**Luciferase and nanoluciferase measurement**. Cells were incubated with ice-cold luciferase lysis buffer (25 mM Tris-HCL pH 7.8, 1% Triton X-100, 15 mM MgSO$_4$, 4 mM EGTA, 1 mM DTT) for 20 min. Both assays were performed in a 96-wells plate format. Luciferase firefly substrate was dissolved in PBS, and consisted of 20 mM Tricine, 2.67 mM MgSO$_4$, 0.1 mM EDTA, 33.3 mM DTT, 0.52 mM ATP, 0.27 mM Acetyl-CoA, 5 mM NaOH, 50 mM MgCO$_3$, and 0.47 mM D-Luciferin (Carl Roth, #CN24.3). For nanoluciferase, the substrate solution included 47.2 $\mu$M Coelenterazine (Carl Roth, #4094.4). Luminescence was detected after 5 min of substrate and lysate co-incubation using the TriStar$^2$ LB 942 Modular Multimode microplate reader (Berthold Technologies).

**High-throughput carrier–carrier adhesion assay**. Met-5A cells were seeded together with Cytodex® 3 microcarrier beads (Sigma Aldrich, #C3275) in a ratio of 500:1, and allowed to adhere and grow for 5 days. Cell-covered beads were then eluted from the culture dish using a 25 mL stripette. The resulting solution was strained through a 100 $\mu$m cell strainer (Corning, #352360). Desiccation shock was induced by placing the bead-containing cell strainer under a running cell-culture

flow hood for 15 min. Afterwards, beads were eluted with assay medium and placed in a cultivation dish coated with HEMA silicate solution (Sigma Aldrich, #P3932), to prevent cell attachment. After indicated time points, carrier complexes were collected by filtering them through a 200 µm cell strainer, which allows individual carriers to pass, while trapping carrier complexes. Adhesions were measured in a high-throughput manner through the use of Met-5A cells stably expressing luciferase (AF23) or nanoluciferase (AF1) using Integra Viaflo pipettes.

**High-throughput carrier-monolayer adhesion assay.** Met-5A cells were seeded on gelatin (Sigma Aldrich, #G1393) coated dishes. Once a confluent monolayer was established, stressed cell-covered beads were seeded onto the monolayer. For the inhibitory experiments, if not indicated otherwise, cells were pretreated with the indicated compound for 30 min in culture medium. Afterwards the adhesion assay was performed as described above.

**Microcarrier labeling.** Cytodex3 microcarrier where stained with either Alexa Fluor 488 NHS Ester (Thermo, A20000) or Alexa Fluor 568 NHS Ester (Thermo; A20003) according to the manufacturer's instructions. Labelled microcarriers where intensively washed with full medium and used as described above.

**Cre-exchange transmission assay.** Met-5A cells stably expressing Cre recombinase (AF32) were seeded on Cytodex® 3 microcarrier beads (Sigma Aldrich, #C3275) and were exposed to desiccation shock as described, and placed on a Met-5A monolayer stably expressing the dTomato-P2A-NanoLuciferase reporter construct (AF34). Nanoluciferase luminescence was measured after 48 h. This represents the initial stress-dependent transmission.

Wild-type Met-5A cells were then seeded on a monolayer, exposed to desiccation shock, and placed together with unstressed cell-covered beads that stably express Cre recombinase (AF32) for 3 h. Carriers were then separated from the monolayer and placed on an unstressed Met-5A monolayer stably expressing the dTomato-P2A-NanoLuciferase reporter construct (AF34). Nanoluciferase luminescence was measured after 48 h. This represents the first stress-independent transmission.

Wild-type cell-covered beads were then exposed to desiccation shock, and seeded on an unstressed wild-type Met-5A monolayer for 3 h. Carriers were then separated from the monolayer and placed on unstressed cell-covered beads stably expressing Cre recombinase (AF32) for 3 h. Carriers were then isolated once more and placed on an unstressed Met-5A monolayer stably expressing the dTomato-P2A-NanoLuciferase reporter construct (AF34). Nanoluciferase luminescence was measured after 48 h. This represents the second stress-independent transmission.

This procedure was continued similarly for the consecutive third and fourth transmissions.

**Adhesion propagation assay.** Met-5A cells stably expressing nanoluciferase (AF1) were seeded on Cytodex® 3 microcarrier beads (Sigma Aldrich, #C3275) and were exposed to desiccation shock as described, and placed on a wild-type Met-5A monolayer. After 24 h, unbound carriers were washed away and nanoluciferase luminescence was measured. This represents the initial stress-dependent transmitted adhesion.

Wild-type Met-5A cells were then seeded on a monolayer, exposed to desiccation shock, and placed together with unstressed cell-covered beads that stably express nanoluciferase (AF1) for 3 h. Carriers were then separated from the monolayer and placed on an unstressed Met-5A wild-type monolayer. After 24 h, unbound beads were washed away and nanoluciferase luminescence was measured. This represents the first stress-independent transmitted adhesion.

Wild-type cell-covered beads were then exposed to desiccation shock, and seeded on an unstressed wild-type Met-5A monolayer for 3 h. Carriers were then separated from the monolayer and placed on unstressed cell-covered beads stably expressing nanoluciferase (AF1) for 3 h. Carriers were then isolated once more and placed on an unstressed Met-5A monolayer stably expressing the dTomato-P2A-NanoLuciferase reporter construct (AF34). Unbound beads were then washed away and nanoluciferase luminescence was measured after 24 h. This represents the second stress-independent transmitted adhesion.

This procedure was continued similarly for the consecutive third and fourth transmissions.

**Spinning disc cell detachment force assay.** The protocol used to measure cell detachment forces was adapted from[36]. In brief, Met-5A cells stably expressing nanoluciferase were seeded on gelatin (Sigma Aldrich, #G1393) coated glass slides. After 2 days, cells were stressed as described above and further grown for another 24 h. Slides were then rotated, lysed, and nanoluciferase activity was measured. For baseline values, cell detachment forces were expressed relative to nanoluciferase values derived from a lysed non-rotated glass slide on which cells were grown to confluence. For carrier-to-monolayer detachment, the same protocol was performed with a Met-5A cell monolayer seeded on gelatin coated glass slides on which Met-5A coated Cytodex® 3 microcarrier beads (Sigma Aldrich, #C3275) stably expressing nanoluciferase were added. To derive meaningful values in terms of generated forces, values were used as described in table 1.1[36], which describes the

relationship between the spinning speed, and the wall shear stress for a range of radial positions across the spinning disc (from the axis of rotation).

**siRNA mediated knockdown.** RNAi-mediated knockdown was performed following the manufacturer's small-interfering RNA (siRNA) transfection protocol (Santa Cruz). In brief, mesothelial cells where transfected with indicated siRNAs on day 3 in cocultivation with microcarriers. In vitro adhesion assay was performed as described above.

**Murine tissue preparation for imaging purposes.** Upon organ excision, organs were fixed overnight at 4 °C in 2% formaldehyde. The next day, fixed tissues were washed three times in Dulbecco's phosphate buffered saline (PBS) (DPBS, GIBCO, #14190-094), and depending on the purpose, either embedded, frozen in OCT compound (Sakura, #4583) and stored at −20 °C, or stored at 4 °C in PBS containing 0.2% gelatin (Sigma Aldrich, #G1393), 0.5% Triton X-100 (Sigma Aldrich, #X100) and 0.01% Thimerosal (Sigma Aldrich, #T8784) (PBS-GT).

**3D imaging of whole-mount tissue samples.** Whole-mount samples were stained and cleared with a modified 3DISCO protocol[37]. In short, samples stored in PBS-GT were incubated with primary antibodies in PBS-GT with shaking, for 36 h at RT. Excessive antibody was removed by thorough washing in PBS-GT for 6–12 h and refreshing the solution every 1–2 h. Incubation with fluorophore-coupled secondary antibodies (Molecular Probes) in PBS-GT for 36 h was followed by thorough washing in PBS-GT as described above. When necessary, samples were dehydrated in an ascending Tetrahydrofuran (Sigma, #186562) series (50%, 70%, 3 × 100%; 60 min each), and subsequently cleared in dichloromethane (Sigma, #270997) for 30 min and eventually immersed in benzyl-ether (Sigma, #108014). Non-cleared samples were imaged in 35 mm glass-bottom dishes (Ibidi, #81218) using a laser scanning confocal microscope (Zeiss LSM710) or SP8 Multiphoton microscope (Leica). Cleared samples were imaged whilst submerged in benzyl-ether with a light-sheet fluorescence microscope (LaVision BioTec).

**3D multiphoton imaging.** For multiphoton imaging, samples were embedded in a 4% NuSieve GTG agarose solution (Lonza, #50080). Imaging was performed using a 25x water-dipping objective (HC IRAPO L 25×/1.00 W) coupled to a tunable pulsed laser (Spectra Physics, Insight DS+). Multiphoton excited images were recorded with external, non-descanned hybrid photo detectors (HyDs). Following band pass (BP) filters were used for detection: HC 405/150 BP for DAPI/Hoechst and Second Harmonic Generation (SHG), a ET 525/50 BP for green channel, 585/40 BP for red channel and a 650/50 BP (magenta) for far red. Tiles were merged using Leica Application suite X (v3.3.0, Leica) with smooth overlap blending and data were visualized with Imaris software (v9.1, Bitplane).

**3D light-sheet imaging.** Whilst submerged in benzyl-ether, specimens were illuminated on two sides by a planar light-sheet using a white-light laser (SuperK Extreme EXW-9; NKT Photonics). EdU and PDPN were excited at 640/30 and 576/23 nm, respectively, and the emitted light was detected using 690/50 and 620/31 nm filters. Optical sections were recorded by moving the specimen chamber vertically at 5-mm steps through the laser light-sheet. Three-dimensional reconstructions were obtained using Imaris imaging software (v9.1, Bitplane).

**Scanning electron microscopy.** Met-5A cells were seeded on Cytodex® 3 microcarrier beads (Sigma Aldrich, #C3275) and exposed to desiccation shock as described. Cell-covered beads were then added to Met-5A cells seeded on gelatin (Sigma Aldrich, #G1393) coated glass slides, and after the indicated time points glass slides were fixed O/N at 4 °C using 3% glutaraldehyde and 0.1% sodium cacodylate buffer, pH 7.4 (Electron Microscopy Sciences, #16538). For animal tissues, adhesions were induced in mice as described, and sacrificed 16 h later. The peritoneum was then fixed in the same manner as the glass slide samples. Samples were dehydrated in a serial dilution of ethanol and dried by the critical-point method, using $CO_2$ as the transitional fluid (Polaron Critical-Point Dryer CPC E3000; Quorum Technologies, Ringmer, UK). Samples were sputter coated with a 7 nm layer of platinum by a sputtering device (Emitech K575; Quorum Technologies) and observed by scanning electron microscopy (JSM 6300F; JEOL, Eching, Germany).

**3D reflectance confocal imaging.** Reflection microscopy images were recorded with a Leica SP8 microscope using a green solid state laser (552 nm). Confocal images were achieved with a pinhole of 44.7 µm and reflectance signal was collected from 547 to 555 nm using a PMT (Hamamatsu R 9624). Z-stacks (intervals of 1 µm) were recorded every 5 min. Images were processed with LAS X (Leica; v3.6.0) and Imaris (Bitplane; v9.3.0) and brightness and contrast were adjusted for optimal visualization.

**2D imaging of murine and human tissue sections.** Fixed tissues were embedded in optimal cutting temperature (OCT) and cut with a Microm HM 525 (Thermo Scientific). Sagittal cross-sections of 7 µm were used for analyses. In short, sections

were fixed in ice-cold acetone for 5 min at −20 °C, and then washed with PBS. Sections were then blocked for non-specific binding with 1% BSA and 5% goat serum in PBS for 60 min at the room temperature, and then incubated with primary antibody in 1% BSA and 5% goat serum in PBS, O/N at 4 °C. The next day, following washing, sections were incubated in PBS with fluorescent secondary antibody, for 120 min at RT. Finally, sections were washed and incubated with Hoechst 33342 nucleic acid stain (Invitrogen, #H1399), washed in ddH$_2$O, mounted with Fluoromount-G® (Southern Biotech, #0100-01), and stored at 4 °C in the dark.

**Masson's trichrome staining**. To visualize deposited matrix, Masson's trichrome staining was performed (Sigma Aldrich, #HT15). In brief, samples were fixed for 10 min in ice-cold acetone at −20 °C, and subsequently washed in dH$_2$O for 5 min. Then, samples were incubated overnight in Bouin's solution (Sigma Aldrich, #HT10132) at the room temperature, and washed the next day under running tap water for 5 min. Samples were then immersed in Weigert's iron hematoxylin solution (Sigma Aldrich, #HT1079) for 3 min, and again washed under running tap water for 5 min. Samples were incubated with Briebrich Scarlet-Acid Fuchsin solution for 5 min, rinsed in dH$_2$O, and incubated with Phosphotungstic/Phosphomolybdic acid solution for 5 min. Finally, samples were immersed in Aniline Blue solution for 10 min, washed in 1% acetic for 2 min, and further washed with dH$_2$O, and then dehydrated through an ethanol gradient. Samples were then dipped 8–10 times and cleared in Roti®-Histol (Carl Roth, #6640) and mounted with Roti®-Histokitt (Carl Roth, #6638).

**Imaging of Met-5A covered carrier–carrier complexes**. Carrier–carrier complexes where fixed in 4% PFA in PBS for 20 min at RT. Afterwards, complexes were washed two times with PBS. Cells were permeabilised for 10 min in 0.1% Triton X-100 (Sigma Aldrich, #X100) in PBS at 4 °C, after which they were washed two times with 0.02% Tween-20 (Sigma Aldrich, #9416) in PBS. Carriers were then blocked for non-specific binding with 5% BSA and 0.02% Tween-20 in PBS for 60 min at 4 °C on a rocking platform, and then incubated with primary antibody in 0.02% Tween-20 in PBS O/N at 4 °C. The next day, following washing, carriers were incubated in PBS with fluorescent secondary antibody for 120 min at 4 °C on a rocking platform. Finally, carriers were washed and incubated with Hoechst 33342 nucleic acid stain (Invitrogen, #H1399).

**Imaging of Met-5A carrier-to-monolayer samples**. For carrier-to-monolayer samples Met-5A cells where seeded on gelatin (Sigma Aldrich, #G1393) coated glass slides. Samples were fixed with 4% PFA in PBS for 20 min at RT. Cells were permeabilised for 10 min in 0.1% Triton X-100 (Sigma Aldrich, #X100) in PBS at 4 °C, after which they were washed two times with 0.02% Tween-20 (Sigma Aldrich, #9416) in PBS. Slides were then blocked for non-specific binding with 5% BSA and 0.02% Tween-20 in PBS for 60 min at 4 °C, and then incubated with primary antibody in 0.02% Tween-20 in PBS O/N at 4 °C. The next day, following washing, carriers were incubated in PBS with fluorescent secondary antibody for 120 min at 4 °C. Finally, slides were washed and incubated with Hoechst 33342 nucleic acid stain (Invitrogen, #H1399).

**Membrane dye labeling of microcarrier and monolayer cultures**. Met-5A cells were labeled with DiO staining solution (Invitrogen, #V22886) according to the manufacturer's instructions, and seeded as a monolayer on gelatin (Sigma Aldrich, #G1393) coated glass chambers (Ibidi, #80287). After 3 days of cultivation, a separate population of cells were seeded together with Cytodex® 3 microcarrier beads (Sigma Aldrich, #C3275) and labeled using the PKH26 Red Fluorescent Cell Linker kit (Sigma Aldrich, #MINI26-1KT). Cell-covered beads were then exposed to desiccation shock as described, and were added to the monolayer culture.

**Microcarrier labelling with primary mesothelial cells**. Cytodex3 microcarriers were stained with either Alexa Fluor 488 NHS Ester (Thermo Scientific, #A20000) or Alexa Fluor 568 NHS Ester (Thermo Scientific, #A20003) according to the manufacturer's instructions.

**Image pre processing**. All image processing and analyses were performed with exported Tif images using Fiji (ImageJ 2.0.0 /1.52c, USA). Fluorescent channels were split and the brightness and contrast were adjusted to reduce background in order to prevent misinterpretation of background as cellular structures in the segmentation step.

**Image pre processing: segmentation**. Mesothelial protrusion analysis was performed using the Advanced weka segmentation Fiji plugin[38]. It utilizes a collection of machine learning algorithms for segmentation. Specifically, pixel-based segmentation is based on image features annotated to different classes. Pixel samples were free drawn and assigned to respective classes, e.g., 'filopodia', 'cell body', or 'background'. Subsequent rounds of training were performed to allot respective pixels and structures to its corresponding classes for improved segmentation. Training features, such as 'Gaussian blur', 'Sobel filter', 'Hessian', 'Difference of Gaussians' and 'Membrane projections' were applied along with default classifier

'Fast Random Forest'. Other settings were kept as default (membrane thickness 1, membrane patch size 19, minimum sigma 1.0, and maximum sigma 16.0). The trained classifier and data were saved for analysis of other 'stressed' and 'unstressed' mesothelial datasets. A macro was written to automate the above steps for other datasets with pause for 5 s (wait(5000) function) after each step for smooth processing. Images from the segmented classes were extracted and subjected to post processing.

**Image post processing: total filopodial surface area**. To quantify the total filopodial surface area, filopodial segments were obtained as described in 'Segmentation'. Brightness and contrast was adjusted from these images, and were converted into binary images. Mean fluorescent intensity was then computed.

**Image post processing: length and width of filopodia**. Length and width of filopodial protrusions were calculated using the ImageJ plugin 'Ridge detection'[39]. In short, filopodial segments were obtained as described in 'Segmentation'. 'Correct position', 'estimate width', 'extend line', 'display results', and 'add to manager' settings were selected. Parameters used included optional parameters (line width: 10, high contrast: 230, low contrast: 87) and mandatory parameters (Sigma: 3.39, lower threshold: 0.51, upper threshold: 1.19, minimum line length: 15.00). Parameters were optimized using preview function for a single dataset and similar values and settings were applied for other datasets. Values of length and width were extracted from the summary tab and exported as an Excel file.

**In vivo EdU labelling**. Animals were injected intraperitoneally with 1 μg of EdU (Invitrogen, #A10044) dissolved in 100 uL PBS on the day of surgery, and sacrificed on day 5. Following organ excision and fixation overnight in 2% formaldehyde, EdU was visualized using the Click-iT™ EdU Alexa Fluor™ 647 imaging kit (Invitrogen, #C10340), according to the manufacturer's instructions. The Click-iT® reaction cocktail was incubated with the samples for 36 h at the room temperature to allow sufficient penetration into the tissue. Tissues were then further processed according to the whole-mount imaging protocol (see '3D imaging of whole-mount tissue samples').

**Localized treatment with topical cellulose**. Small-molecule inhibitors were solubilized in sterile 2% hydroxyethyl-cellulose (Sigma Aldrich, #09368). Compounds were being added immediately prior to surgery, and were derived from a 100–150 mM stock solution to minimize the final DMSO content. The final solution (200 μL per 30 g body weight) was sandwiched between the visceral and parietal layer of the injured cecum and peritoneum respectively.

**Animals**. All animal experiments were conducted under strict governmental and European guidelines and were approved by the local government for the administrative region of Upper Bavaria, under license number 55.2-1-54-2532-150-2015. Pathogen-free male and female C57BL/6 mice (6–10 week old) were obtained from Charles River and group-housed in climate-controlled quarters with a 12 h/12 h light/dark cycle. Animals were allowed food and water ad libitum.

Rosa26$^{mTmG}$ or Rosa26$^{tm1(DTA)Lky}$ crossed with Procr$^{CreERT2-IRES-tdTomato}$ mice were used in this study (Jackson Laboratories). For lineage-tracing experiments induced in 6–8 week adult mice, animals received three intraperitoneal injections of (Z)-4-Hydrotamoxifen (Sigma Aldrich, #H7904, 2 mg per 25 g body weight, diluted in 100 μL corn oil (Sigma Aldrich, #C8267)) every other day to induce activation of Cre recombinase. For selective ablation of PROCR+ mesothelial cells, animals received a single administration of (Z)-4-Hydrotamoxifen (Sigma Aldrich, #H7904, 2 mg per 25 g body weight, diluted in 100 μL corn oil (Sigma Aldrich, #C8267)) immediately after surgery before closing the abdomen.

**Murine adhesion model**. Mice were anesthetized by an intraperitoneal injection of a Medetomidin (500 μg/kg), Midazolam (5 mg/kg) and Fentanyl (50 μg/kg) cocktail, hereafter referred to as MMF. Monitoring anesthetic depth was assessed by toe reflex. Eyes were covered with Bepanthen to avoid dehydration, and the abdomen was shaved and disinfected with betadine and sterile PBS. Animals were kept on their backs on a heating plate at 37 °C. A midline laparotomy (1–1.5 cm) was performed through the skin and peritoneum. Four hooks, positioned around the incision and fixed to a retractor and magnetic base plate, allowed for clear access to the abdominal cavity. A small surgical brush was used to gently abrade the peritoneal surface and apposing cecal surface. Two surgical knots using 4-0 silk sutures (Ethicon) were then placed through the serosal surface of the peritoneum. A cotton swab was used to gently apply a dab of talc powder (Sigma Aldrich, #243604) onto the injured surfaces. Before closure of the incision, buprenorphine (0.1 mg/kg) was pipetted in the abdomen to allow for initial postsurgical analgesia. For long-term analgesia, metamizol (Novalgin, 200 mg/kg) was provided through daily injection. The peritoneum and skin were closed with two separate 4-0 silk sutures (Ethicon). Upon closure of the incision, mice were woken up by antagonizing the MMF solution through a subcutaneous cocktail injection of Atipamezol (1 mg/kg) and Flumazenil (0.25 mg/kg). Mice were allowed to recover on a heating pad, after which they were housed together (females) or individually (males), and followed

for 1–5 days. Adhesions were scored using gross morphological features that indicated adhesion development. Five individual adhesion features were scored (see Supplementary Table 1) that together provided a cumulative value that determined the total adhesion score. With this system, complete absence of adhesions was scored as 0, whereas the maximum adhesion score was 15.

**Human tissue.** All human samples have been obtained during surgery at the Department of Surgery, Klinikum rechts der Isar, Technical University of Munich, following approval of the local ethics committee of the Technical University of Munich, Germany (Nr. 173/18 S). Adhesions were intraoperatively diagnosed and dissected from the respective organs and prepared for further analysis. Informed consent was obtained from all subjects after surgery.

**Single-cell RNA sequencing (Drop-seq).** Met-5A cells were grown in culture and stressed for 15 min by desiccation. Afterwards, at the indicated time points, samples were incubated for 5 min in Trypsin-EDTA solution at 37 °C. Trypsin was inactivated with ice-cold assay medium and cells were washed twice with ice-cold PBS. Drop-seq experiments were performed as described previously[20,21], with few adaptations during the single-cell library preparation. Briefly, single cells were diluted in PBS, supplemented with 0.04% bovine serum albumin up to a final concentration of 100 cells/uL. Using a microfluidic PDMS device (Nanoshift), single cells were co-encapsulated in droplets with barcoded beads (Chemgenes Corporation, Wilmington, MA) at a final concentration of 120 beads/uL. Droplets were collected for 15 min/sample. After droplet breakage, beads were harvested, washed, and prepared for on-bead mRNA reverse transcription (Maxima RT, Thermo Fisher). Following an exonuclease I (New England Biolabs) treatment for the removal of unused primers, beads were counted, aliquoted (2000 beads/reaction, equals ~100 cells/reaction), and pre-amplified by 13 PCR cycles (primers, chemistry, and cycle conditions identical to those previously described in Macosko et al.[20]). PCR products were pooled and purified twice using 0.6x clean-up beads (CleanNA). Prior to tagmentation, cDNA samples were loaded on a DNA High Sensitivity Chip on the 2100 Bioanalyzer (Agilent) to ensure transcript integrity, purity, and amount. For each sample, 1 ng of pre-amplified cDNA from an estimated 1000 cells was tagmented by Nextera XT (Illumina) with a custom P5 primer (Integrated DNA Technologies). Single-cell libraries were sequenced in a 100 bp paired-end run on the Illumina HiSeq4000 using 0.2 nM denatured sample and 5% PhiX spike-in. For priming of read 1, 0.5 μM Read1CustSeqB was used (primer sequence: GCCTGTCCGCGGAAGCAGTGGTATCAACGCAGAGTAC).

**Bioinformatic processing of single-cell RNA sequencing data.** The Drop-seq core computational pipeline was used for processing next generation sequencing reads of the scRNA-seq data, as previously described[20]. Briefly, STAR (version 2.5.2a) was used for mapping[40]. Reads were aligned to the hg19 genome reference (provided by Drop-seq group, GSE63269). For barcode filtering, we excluded barcodes with less than 200 genes detected. A high proportion (>10%) of transcript counts derived from mitochondria-encoded genes may indicate low cell quality, and we removed these unqualified cells from the downstream analysis. After obtaining the DGE data matrix, we used Seurat for dimension reduction, clustering and differential gene-expression analysis[20].

**Principal component analysis.** Using only variable genes, a principal component analysis (PCA) was performed. The top 15 principal components were used as input for the Seurat *FindClusters* function at a resolution of 0.5. This method accomplishes a clustering of the cells by embedding them in a graph like structure. A *k*-nearest neighbor graph is used, in which any two cells (represented as nodes) that are connected by an edge have an edge weight that is among the *k* smallest distances from the first node to any other. Thus, edges are drawn between cells with similar gene-expression patterns. Modularity optimization methods such as the Louvain Algorithm try to reveal parts of the graph with different connectivity and therefore divide the graph into separate interconnected modules.

**Partition based graph abstraction method.** To visualize the clustering result of the high dimensional single-cell data, the Fruchterman-Reingold algorithm from the Python toolkit Scanpy was employed[41]. In addition, to display the connectivity between the cell groups the partition based graph abstraction (PAGA) method was used[41]. The cells were grouped according to the time point of extraction. In the graph, those groups are represented as nodes and edges between the nodes show the connectivity or relatedness of these groups, therefore quantifying their similarity with respect to gene-expression differences.

**Time resolved pathway analysis.** To predict the activity of pathways and cellular functions based on the observed gene-expression changes, we used the Ingenuity® Pathway Analysis platform (IPA®, QIAGEN Redwood City, www.qiagen.com/ingenuity) as previously described[42]. The analysis uses a suite of algorithms and tools embedded in IPA for inferring and scoring regulator networks upstream of gene-expression data based on a large-scale causal network derived from the Ingenuity Knowledge Base. Using the 'Downstream Effects Analysis'[43] embedded

in IPA we aimed at identifying those biological processes and functions that are likely to be causally affected by upregulated and downregulated genes in the single-cell transcriptomics dataset. In our analysis we considered genes with an overlap *P* value of >7 (log10) that had an activation *Z*-score >2 as activated and those with an activation *Z*-score < −2 as inhibited.

**Statistics and reproducibility.** All data represent the mean ± SEM. A Shapiro–Wilk's test ($p > 0.05$) as well as visual inspection of the respective histograms, normal Q–Q plots and box plots were used to test whether samples were normally distributed (approximately), using IBM SPSS Statistics version 23. Two group comparisons were made using an unpaired Student's *t* test for normally distributed data or a Mann–Whitney *U* test as the nonparametric equivalent. Comparisons between three or more groups were performed using a one-way ANOVA followed by Tukey's post hoc test for normally distributed data, or with a Kruskal–Wallis *H* test for non-normally distributed data. A value of $p < 0.05$ was considered statistically significant, where $*p < 0.05$, $**p < 0.01$, and $***p < 0.001$. Analyses were performed with GraphPad Prism version 6 (GraphPad Software, Inc.). Directionality/Polar-coordinates plot was performed using the "ggplot2" library #1234 in R #5678 Version 3.4.1, 2[44,45]. All experiments were repeated at least three times independently with similar results.

**Reporting summary.** Further information on research design is available in the Nature Research Reporting Summary linked to this article.

## Data availability
The source data underlying Figs. 1a, 2a–d, 6d, h and 7c and Supplementary Figs 1a and 5d are provided as a Source Data file.

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

## Acknowledgements

We thank Dr. Steffen Dietzel and the Core Facility Bioimaging at the Biomedical Centre of the Ludwig-Maximilians-Universität München for access and support with the multiphoton system. We also thank Dr. Reinhardt Faessler for allowing us to use the spinning disc device, and Gabriele Mettenleiter and Aydan Sardogan for technical assistance.

## Author contributions

Y.R., A.F., and T.K. designed and oversaw the study. A.F. and T.K. performed the bulk of the experiments and data analysis. Y.R., A.F., and T.K. wrote the manuscript. T.K. prepared all figures. P.R., S.C., M.S., Me.A. performed experiments and data analysis. J.W. wrote all animal work related amendments and gave decisive input for animal related experiments. P.N. provided human patient material. H.B.S., F.J.T., A.F., A.W., Mi.A., K.S., and K.H. provided expertise in experimental set-up and data interpretation.

## Competing interests

The authors declare no competing interests.
