## [Peer Review File · Nature Communications]

Reviewers' Comments:

Reviewer #1:

Remarks to the Author:

This study explores poorly understood mechanisms of post-surgical adhesions between organ surfaces. The authors examine interactions between human mesothelial cells growing on beads and on the culture plate and report that subjecting the cells to air or talcum powder (both linked to triggering post-surgical adhesions) results in a strong adhesion between cell monolayers.

"Stressed" Met-5A cells on Matrigel formed dynamic assemblages of membrane protrusions reaching from one cell to another. The authors show that under some conditions the adhesion is accompanied by cell-cell fusion, and, using single-cell transcriptomics, demonstrate that the development of the strong adhesion involves certain cytoskeletal and calcium effectors.

Treatments targeting the cytoskeletal and calcium pathways that are linked to the adhesion in the new in vitro model prevent adhesions in a preclinical mouse model. While the new model and many of the results are interesting and exciting, the paper is very hard to read. In many cases it was very difficult for me to understand the details of the experiments and I had to guess what exactly was done. Furthermore, it appears that in most of the paper the authors use the terms 'fusion' and 'fusing' and 'fusogenic' when, in the absence of direct evidence for cell fusion, they actually mean 'adhesion' and propensity of the cell monolayers to generate the strong adhesions between cell monolayers. While the authors present some evidence that the adhesion between cell monolayers is accompanied by cell-cell fusion (Figure 2F,H, Video S5), they start to discuss adhesion as fusion before showing any evidence for fusion. Both adhesion and fusion are expected to depend on actin cytoskeleton and calcium signaling. However, these are quite different processes and I found no evidence that the strong adhesion between the cells in the presented experiments can be explained by fusion. While I cannot fully evaluate the paper as presented, it may suggest new and interesting mechanistic insights into a very serious medical problem.

Specific comments.

1) I suggest the authors should go through all the figures and make sure that all the details of the presented experiments are fully explained in the Figure legends and/or clearly identifiable parts of the Methods. For instance, I found no explanation of the design of the experiments presented in Fig. 1b, c, d. Why in Figures 1b, c an increase in nanoluc (=nanoluciferase) emission is taken as a measure of carrier-to-monolayer aggregation? Does the measurement characterize the delivery of the nanoluc cells into the plane of the flat monolayer? At what time point are the measurements in b and c carried out? Why and how the aggregation is quantified by counting bound carriers in d? Do the counted carriers represent single beads or aggregates of the beads? The Subsection named '3D reflectance confocal imaging' contains only "Insert text here." I have not found the specific description of "stressing" treatments in the experiments with Met-5A cells plated at the Matrigel (desiccation? talcum?). I have not found in the Methods for how long the cells in the monolayer or on the beads were subjected to desiccation in preparation for single-cell RNA-seq.

2) In Fig. 1 and in the discussion of this figure and throughout the paper the authors seem to use the term 'fusion' as a synonym for 'adhesion'. This is especially confusing in the paper that also presents data on cell fusion (Figure 2F,H, Video S5). In the subsection 'Pathway analysis identifies the key fusogenic steps in adhesion formation' the authors discuss the fusogenic steps, fusogenic cell markers, fusogenic cell phenotypes but show no cell fusion data that would substantiate the link between the changes in the transcriptional program and fusion.

3) The images presented in Fig. 2F, H indicate the exchange of cytoplasmic content during early stages of adhesions. I have not found at what time after the "stress" (desiccation?) were the images taken and thus cannot evaluate the statement that the exchange of cytoplasmic content takes place "during early stages of adhesions". Do the authors observe formation of multinucleated cells? Can the exchange proceed via nanotubes? Can the cell dissociation and lifting (as in the preparation for single-cell RNA-seq) be used to quantitatively characterize formation of the multinucleated cells?

4) I have had difficulties in understanding the experiment presented in Fig 2I. Should the sentence "To expand on these findings, we mixed stress-exposed carrier-bound cells with unstressed

carrier-bound cells, and labeled them with red or green membrane lipid dyes respectively.” be turned around to reflect that the stressed carrier-bound cells and unstressed carrier-bound cells were first labeled and then mixed? Then, I do not understand why the beads with the stressed cells (red) show very low CD44 expression (much lower than that for the beads with unstressed cells). Also, while I do see green cells on the stressed (red) carriers, at this very low magnification I cannot identify any cells co-labeled with green and red probes and, thus, see no evidence of “active membrane mixing with an adhesion phenotype”.

5) Considering that carrier-to-monolayer aggregation develops as fast as 60 min after injury, why are the changes in the expression of RNAs in the Met-5A cells (single-cell RNAseq) observed only starting from 8h? Can the transcriptional changes detected at 8 hours post-stress reflect the consequences of cell-cell adhesion rather than the processes leading to the adhesion?

6) Cell-cell adhesion is known to depend on both calcium signaling and actin remodeling pathways. Thus, I do not see how finding that strong adhesive interactions between mesothelial cells subjected to desiccation involve cytoskeletal and calcium effectors argues for the importance of the fusogenic pathways.

7) Fig. S1 shows that stress and clustering of the carrier aggregates is accompanied by a very impressive change in the expression of the adhesion proteins such as CD44 in Met-5A cells. The analysis is carried out 2 days after the stress and the change in the expression is discussed as “bead clustering induces biomarker expression”. It would be interesting to test whether the change in the adhesion protein expression precedes or follows the bead clustering.

8) Is drastic increase in carrier-to-monolayer aggregation for the Met-5A cells subjected to air or talcum desiccation specific for mesothelial cells or can be observed for other cells, say fibroblasts?

Minor comments

1) Please number the figures.

2) Please format the reference: “Jonathan M Tsai, et al (2018). Surgical adhesions in mice are derived from mesothelial cells and can be targeted by antibodies against mesothelial markers. *Sci. Transl. Med.* 10, ean6735.”

Reviewer #2:

Remarks to the Author:

The authors provide fundamental advances from current existing knowledge, into contribution of mesothelial cells to adhesive development. The authors provide comprehensive in vitro studies examining histologic imaging in conjunction with molecular biologic analysis of the pathophysiology of adhesions, along with complementary in vitro studies which also define the timeline of adhesion development.

While demonstrating mesothelial cell contribution to adhesion, the authors may be overstating the singular existence of this pathway, and fail to indicate possible implication of fibroblast contribution. The authors do not assess fibroblast contribution in their in vitro models, and have chosen an abrasion in vivo in which “islands” of mesothelial cells may persist. It would be interesting to examine similar endpoints in a peritoneal excision model, to assess potential dual contributions of mesothelial cells and fibroblasts.

The authors fail to include among their references many prior reports which have also identified the contribution of mesothelial cells to adhesion formation.

Specific minor comments are:

- While not trying to diminish the potentially severe consequences of adhesions, several statements in the introduction are phrased in misleading ways, making the consequences worse than generally accepted. As an example, three-fourths of abdominal operations have adhesions

obstruct the small bowel; occurrence is that high, but "obstruction" is much lower. Similarly, a third of patients being readmitted is four years; the SCAR study looked at a 10 year window.

- The Tsai reference is not provided

Reviewer #3:

Remarks to the Author:

Review on the manuscript

by Fischer et al.,

"Post-surgical adhesions are caused by membrane bridges and fusions between mesothelial surfaces"

Post-surgical adhesions are still an unmet challenge causing intense morbidity and mortality. Recently, Yuval Rinkevich's lab identified the mesenchymal cell lineage which contributes to the formation of post-surgical adhesions. In the here reviewed manuscript, he and his collaborators now uncovered the underlying mechanisms and how to therapeutically address the currently unresolved problem of post-surgical adhesions. The authors developed a novel in vitro assay to visually monitor the early interactions between organ surfaces in real time and in response to distinct pathogenic factors. If pathogenic factors are applied in this in vitro model of mesothelial cells covered microcarriers, adhesion to an unstressed monolayer of mesothelial cells was instantly initiated resulting in strong adhesion of the Met5A cells grown on the carriers and the underlying monolayer. This process follows a sequence of events, in early phases post injury, proliferation of mesothelial cell occurred, thereafter markers of EMT (epithelial-mesenchymal - transition) were detected including WT1, α -SMA, indicative of a contractile cytoskeleton. This is followed by matrix deposition with fibronectin and collagen type III. To validate these in vitro results, the authors developed a valuable murine model closely mimicking adhesion as occurring in humans. Results originating from this preclinical murine model confirm the results of the in vitro model. Most interestingly, RNA seq analysis assessed an unbiased gene expression, and pathway analysis, identified genes regulating calcium homeostasis and cytoskeleton adaptation. were mainly up-regulated. This, most interestingly, could be confirmed in post-surgical adhesions from human patients. Small molecules interfering either with calcium influx or as with cytoskeleton gene expression after adhesion stimulation could very convincingly, avoid adhesion formation, this is a clear breakthrough in clinical science, and as such I appreciate this manuscript very much.

Major comments

- 1) I would recommend to the authors to describe in more detail why the mesothelial cell line is appropriate to reflect primary cell lines.
- 2) I am convinced of the therapeutical approach the authors have chosen, and thereby provide a new avenue of treating post-surgical adhesions. As a scientist though, given they have the candidates, I would be even more enthusiastic, if they choose a complementary genetic approach to silence candidate genes in mesothelial cells and thereafter assess the fusion and adhesive capacity in vitro.

Reviewer #4:

Remarks to the Author:

This manuscript describes a follow-up study on the recent paper by authors (Tsai et al., Sci Transl Med. 2018) that examines early cellular and molecular events that produce pathological mesothelial adhesions. Tsai et al. showed that the earliest cellular event in adhesion formation is the fusion between stress-activated mesothelial cells. Current study builds upon Tsai et al. and identified further aspects of mesothelial cell fusion. Intriguingly, using RNA-seq in an in vitro system with human mesothelial cell line, they identify early stress-response genes - cytoskeletal

and calcium effectors. By using an available panel of small molecule modifiers of their top target genes, authors show that they can reduce fusogenic activity of cells in vitro with 4 small molecules and that they have anti-adhesion properties in vivo, in the mouse model.

Despite some intriguing, potentially therapeutically viable leads that the study identifies, there are numerous concerns with: (a) mechanistic depth of the study, (b) artificial nature of the in vitro system and (c) rigor.

1) All of the major findings in this manuscript are derived from an in vitro system using Met-5A human mesothelial cell line. Met-5A cell line is derived in late 1980s by pRSV-T plasmid-induced transformation of mesothelial cells isolated from pleural fluids of non-cancerous individuals. This is a very significant concern. Other than being immortalized, the cell line is specific to pleural cavity, rather than peritoneal cavity. While authors make many extrapolations of their in vitro data on Met-5A cells to peritoneal mesothelium in vivo, it remains unclear just how similar or different pleural and peritoneal mesothelial cells are (even without immortalization).

It is more than likely that many behaviors that Met-5A cells display in vitro are not reflective of the normal and pathological behaviors by peritoneal mesothelial cells in vivo (even though some of the molecular events are suggested to be conserved).

Further, in an in vitro system, Met-5A cells are cultured out of the normal context of peritoneum and all other cell types (immune, fibroblasts) that likely participate and strongly modulate early-stage response by mesothelium to injury are not present in the in vitro system.

2) In an in vitro model, authors fairly conclusively showed that Met-5A cells undergo true cell fusion with cytoplasmic exchange early after stress. However, the molecular mechanism of fusion is not truly studied. The early response genes (cytoskeletal and calcium effector) that authors identify via single-cell RNA-seq on Met-5A cells and immunostaining, can be important for activating cellular protrusion-making activity by Met-5As, rather than cell fusion process by the protrusions per se. Further, authors document multiple morphological types of cellular protrusions that stressed Met-5As form, but they did not study if certain (or all) protrusion types initiate cell fusion and how.

Generally, despite identifying some certainly interesting fusion-promoting gene targets, the molecular mechanism of the fusion process is not disclosed.

3) Authors make strong claims regarding anti-adhesion potential of Bepriidil (calcium channel blocker), Rhosin (Rho-GEF inhibitor), CK-666 (Arp2/3 complex and actin assembly inhibitor), and Golgicide A (Arf1-mediated actin organization inhibitor). These claims are based on in vitro data on Met-5A cells (Fig 4a-c) and on limited in vivo studies in mice (Fig 4d). This is certainly a superficial and preliminary assessment and in order to truly establish any given gene as a critical target in mesothelial fusogenesis, a more comprehensive study, involving genetic mouse model(s) and gene perturbation studies in vitro in primary peritoneal mesothelial cells (both loss- and gain-of-function) will be necessary. At present, small molecule studies on Bepriidil, Rhosin, CK-666 and Golgicide A are fairly preliminary, despite being intriguing.

4) Study rigor. Although likely addressable, at present manuscript lacks replicate "n" values for all experiments throughout. Statistical analysis is rudimentary (p values are shown on figures, but underlying data is not well defined).

Original decision letter from the Editor

Dear Dr. Rinkevich,

Your manuscript entitled "Post-surgical adhesions are caused by membrane bridges and fusions between mesothelial surfaces" has now been seen by 4 referees, whose comments are appended below. You will see from their comments copied below that while they find your work of considerable potential interest, they have raised quite substantial concerns that must be addressed. In light of these comments, we cannot accept the manuscript for publication, but would be interested in considering a revised version that addresses these serious concerns.

In particular, we would like to address the concerns of reviewer 1 on fusion, those of reviewers 2 and 3 by providing more mechanistic insight on the fusion process, and to convincingly address the concerns of reviewer 3 on relevance of the system. It will also be essential to address the concern of referee 4 on the cell line used. This is not to say that we consider the other concerns of our referees any less important.

Author's response

We have carefully addressed each of the reviewer's points and have provided detailed answers as you will find outlined below. The fusion aspect has been addressed textually. As we cannot definitively rule out that membrane bridging events between mesothelial cells progress to adhesions without complete cell fusions, we instead refer to these events as 'adhesions' to cover all aspects of mesothelial cell-cell bridging, whether complete or partial fusions. To prove the physiologic relevance of our novel in vitro assay, we have now isolated primary abdominal mesothelial cells and exposed them to desiccation stress under same culturing conditions as in our in vitro assay. Under these conditions we observed: 1) development of expansive membrane protrusions, 2) rapid bead clustering or adherence of beads to a monolayer culture, and 3) same pharmacologic targets that were originally identified in our in vitro assay as preventative agents, completely inhibit adhesions in primary abdominal mesothelial cells. We conclude from these experiments that our discovered novel mechanism are generalizable to mesothelial surfaces of various organs. We have also conducted new in vivo genetic lineage tracing experiments to address the potential contributing role of fibroblasts to adhesions. Here, we find that fibroblastic cells within the adhesion core originate from a mesothelial source. Further, we use new in vivo genetic methods to specifically deplete mesothelial surfaces prior to adhesion stimulus and report that adhesions completely fail to develop in the absence of mesothelial cells, thus making fibroblasts dispensable in the adhesion process. We have now also included new mechanistic studies to show that transcriptional silencing of core adhesion-inducing genes mimics the pharmacologic phenotypes seen with our small molecules and inhibit adhesion formation.

We hope you will find the referees' comments useful as you decide how to proceed. Should further experimental data or analysis allow you to address these criticisms, we would be happy to look at a substantially revised manuscript. However, please bear in mind that we will be reluctant to approach the referees again in the absence of major revisions. If the revision process takes significantly longer than three

months, we will be happy to reconsider your paper at a later date, as long as nothing similar has been accepted for publication at Nature Communications or published elsewhere in the meantime.

We are committed to providing a fair and constructive peer-review process. Do not hesitate to contact us if you wish to discuss the revision or if there are specific requests from the reviewers that you believe are technically impossible or unlikely to yield a meaningful outcome.

When resubmitting your paper, please highlight all changes in the manuscript text file. We also ask that you ensure that your manuscript complies with our editorial policies. Specifically, please ensure that the following requirements are met, and any relevant checklists are completed or updated and uploaded as a Related Manuscript file type with the revised article:

Author's response

All of our textual changes have been highlighted through Track Changes and we have ensured compliance with the editorial policies.

Reviewer's comments:

Reviewer #1 (Remarks to the Author):

This study explores poorly understood mechanisms of post-surgical adhesions between organ surfaces. The authors examine interactions between human mesothelial cells growing on beads and on the culture plate and report that subjecting the cells to air or talcum powder (both linked to triggering post-surgical adhesions) results in a strong adhesion between cell monolayers. "Stressed" Met-5A cells on Matrigel formed dynamic assemblages of membrane protrusions reaching from one cell to another. The authors show that under some conditions the adhesion is accompanied by cell-cell fusion, and, using single-cell transcriptomics, demonstrate that the development of the strong adhesion involves certain cytoskeletal and calcium effectors. Treatments targeting the cytoskeletal and calcium pathways that are linked to the adhesion in the new in vitro model prevent adhesions in a preclinical mouse model. While the new model and many of the results are interesting and exciting, the paper is very hard to read. In many cases it was very difficult for me to understand the details of the experiments and I had to guess what exactly was done.

Furthermore, it appears that in most of the paper the authors use the terms 'fusion' and 'fusing' and 'fusogenic' when, in the absence of direct evidence for cell fusion, they actually mean 'adhesion' and propensity of the cell monolayers to generate the strong adhesions between cell monolayers. While the authors present some evidence that the adhesion between cell monolayers is accompanied by cell-cell fusion (Figure 2F,H, Video S5), they start to discuss adhesion as fusion before showing any evidence for fusion. Both adhesion and fusion are expected to depend on actin cytoskeleton and calcium signaling. However, these are quite different processes and I found no evidence that the strong adhesion between the cells in the presented experiments can be explained by fusion. While I cannot fully evaluate the paper as presented, it may suggest new and interesting mechanistic insights into a very serious medical problem.

Author's response

We thank the reviewer for his comments. Although the disease is termed 'surgical adhesions' its nomenclature is a clinical terminology stemming from observations of physical attachments between organ surfaces, and is not a biological terminology to say cells attach to one another. Showing physical cellular adhesions with actin cytoskeleton and calcium signaling driving the clinical manifestations of this disease, refigures the current and long-standing etiology of this disease which is wound-derived extracellular matrix deposition at sites of surface injury as driving this welding phenomenon between organ surfaces.

We also agree that the applied techniques can be confusing for the uninitiated reader, and have therefore significantly improved our phrasing to address this concern, and have now included new schematics where appropriate. You can find these changes highlighted through **Track Changes**. More specifically, we have also changed our phrasing with regards to fusion. Although we believe that cell fusions, or more appropriately, membrane fusions, constitute an important step in the adhesion process, we cannot definitively rule out that cell-to-cell membrane attachments without complete fusions could also play a role in surgical adhesion development, e.g. through physical attachments of interconnecting protrusions, or

through adherence and junctional protein-mediated cell-cell binding. Nonetheless, we have decided to omit the term fusion where it was not scientifically appropriate, and have instead used the term 'adhesion' to refer to these events.

Specific comments.

1) I suggest the authors should go through all the figures and make sure that all the details of the presented experiments are fully explained in the Figure legends and/or clearly identifiable parts of the Methods. For instance, I found no explanation of the design of the experiments presented in Fig. 1b, c, d. Why in Figures 1b, c an increase in nanoluc (=nanoluciferase) emission is taken as a measure of carrier-to-monolayer aggregation? Does the measurement characterize the delivery of the nanoluc cells into the plane of the flat monolayer? At what time point are the measurements in b and c carried out? Why and how the aggregation is quantified by counting bound carriers in d? Do the counted carriers represent single beads or aggregates of the beads?

Author's response

We agree that the use of both assays can be confusing without the textual support. The 'adhered carrier assay' simply looks at the number of carriers that have physically adhered to a monolayer of Met-5A cells, after thorough washing to remove unbound carriers. We have included this assay because it actually addresses binding, rather than addressing it by proxy (e.g. through nanoluciferase emission). We have now included representative images with adhered carriers to visually show these bindings phenotypes and improve the clarity of the method (Figure 1B in the revised manuscript). In Figure 1C and D we have switched to a nanoluciferase assay, as this allows for the measurement of adhesions in a more high-throughput setting. For example, our FDA screen could never have been carried out without this assay (Figure 5A in the revised manuscript). Based on the reviewer's comment, we have now included a schematic of the nanoluciferase assay in Extended Figure 1 to provide more insight into the details of the technique and improve readability of the manuscript.

The Subsection named '3D reflectance confocal imaging' contains only "Insert text here."

Author's response

We apologize for this and have included the appropriate segment.

I have not found the specific description of "stressing" treatments in the experiments with Met-5A cells plated at the Matrigel (desiccation? talcum?).

Author's response

These cells have been stressed by desiccation. We have now added this to the textual segment.

I have not found in the Methods for how long the cells in the monolayer or on the beads were subjected to desiccation in preparation for single-cell RNA-seq.

Author's response

These cells have been stressed by desiccation for 15 min. We have added this information to the schematic in Extended Figure 6A, and updated the figure legend.

2) In Fig. 1 and in the discussion of this figure and throughout the paper the authors seem to use the term 'fusion' as a synonym for 'adhesion'. This is especially confusing in the paper that also presents data on cell fusion (Figure 2F,H, Video S5). In the subsection 'Pathway analysis identifies the key fusogenic steps in adhesion formation' the authors discuss the fusogenic steps, fusogenic cell markers, fusogenic cell phenotypes but show no cell fusion data that would substantiate the link between the changes in the transcriptional program and fusion.

Author's response

As mentioned above, we have refrained from using fusion where not appropriate, and have limited this term for the events represented in Figure 4.

3) The images presented in Fig. 2F, H indicate the exchange of cytoplasmic content during early stages of adhesions. I have not found at what time after the "stress" (desiccation?) were the images taken and thus cannot evaluate the statement that the exchange of cytoplasmic content takes place "during early stages of adhesions". Do the authors observe formation of multinucleated cells? Can the exchange proceed via nanotubes? Can the cell dissociation and lifting (as in the preparation for single-cell RNA-seq) be used to quantitatively characterize formation of the multinucleated cells?

Author's response

The events depicted in Figure 2F (now Figure 3A) were captured 24 hours after desiccation stress. For Figure 2G (now Figure 3B) cells were transferred 24 hours after stress (initial stress-dependent transmission), and each subsequent (stress-independent) transmission was performed with 3 hour intervals. For Figure 2G (now Figure 3C) we imaged cells after 24 hours. The methodological details of these experiments have now been added to the text. We agree that these time points do not necessarily represent 'early stages', and have therefore changed the text accordingly. We have never observed any indication of multinucleated cells and believe fusion is limited to the protrusion network and cytoplasmic exchange, possibly via nanotubes, without the complete merging of cell bodies. We added a sentence to help clarify this. A detailed investigation into the working mechanisms of this protrusion-mediated fusion would necessitate a host of additional experiments (e.g. super-resolution microscopy) that we feel would go beyond the scope of this paper.

4) I have had difficulties in understanding the experiment presented in Fig 2I. Should the sentence "To expand on these findings, we mixed stress-exposed carrier-bound cells with unstressed carrier-bound cells, and labeled them with red or green membrane lipid dyes respectively." be turned around to reflect that the stressed carrier-bound cells and unstressed carrier-bound cells were first labeled and then mixed? Then, I do not understand why the beads with the stressed cells (red) show very low CD44 expression (much lower than that for the beads with unstressed cells). Also, while I do see green cells on the stressed (red) carriers, at this very low magnification I cannot identify any cells co-labeled with green and red probes and, thus, see no evidence of "active membrane mixing with an adhesion phenotype".

Author's response

You are correct in your assumption that both cell-covered carriers were first labelled and then mixed. We agree that the CD44 protein expression in the stressed (red colored) cells show a lower protein expression of CD44 compared to the other carriers, but we would like to stress that under uninjured conditions, cells show no CD44 protein expression at all (see Extended Figure 1C). Thus, both red and green carriers are clearly stressed, with variation in their CD44 protein expression. We have other examples (see Figure 1 below) where both types of carriers show strong protein expression of CD44, but for these examples the lipid dye coverage in the original cells prior to their mixing was low, thus precluding them from incorporation into the main manuscript. In all cases, we see clear examples of cells with both red and green labelling (see also the white arrow in Figure 1 below), supporting the active mixing of membranes. We have included a zoomed-in image in the manuscript to help clarify this point.

Figure 1.

5) Considering that carrier-to-monolayer aggregation develops as fast as 60 min after injury, why are the changes in the expression of RNAs in the Met-5A cells (single-cell RNAseq) observed only starting from 8h?

Author's response

We expect that many of the key genes identified in this manuscript act rapidly, most likely before the complete 8 hour time point, and preceding actual adhesion formation. We agree that inclusion of an earlier time point for our RNAseq analyses would allow a better characterization of the dynamic gene expressions at early onset of disease. However, considering these experiments would require a significant waiting time, and would not add to the existing novel findings, we currently feel it would be beyond the scope of the revisions. Furthermore, it is clear from the principle component analyses that the 8 hour time window differs significantly from the later time points, supporting the idea that early pre-adhesion genes may still be active at this stage.

Can the transcriptional changes detected at 8 hours post-stress reflect the consequences of cell-cell adhesion rather than the processes leading to the adhesion?

Author's response

Mesothelial cells normally express cell-cell adhesion molecules similar to epithelium. Yet these are independent of the adhesion disease, both in vitro and in vivo. In theory it is possible that a part of these genes reflect a consequence of reacquiring an epithelial phenotype with cell-cell adhesion. However we believe these genes are irrelevant to the disease phenotype, and act secondary to adhesion-initiating genes, then the selective targeting of them pharmacologically or through siRNA's would not result in the complete prevention of adhesions. Rather, they would inhibit steps that occur subsequently, e.g. proliferation or differentiation into mesenchymal cells. This is contrary to what we observed.

6) Cell-cell adhesion is known to depend on both calcium signaling and actin remodeling pathways. Thus, I do not see how finding that strong adhesive interactions between mesothelial cells subjected to desiccation involve cytoskeletal and calcium effectors argues for the importance of the fusogenic pathways.

Author's response

We have looked into the possible role of adhesion molecules and their involvement in mediating mesothelial adhesions. Although we observed inhibition of bead clustering in our in vitro assay after blocking various cadherins through blocking antibodies or small interfering RNA's (see Figure 3 below), these results did not match our in vivo observations. In Extended Figure 10 of the revised manuscript we looked into several adhesion proteins that we targeted through blocking antibodies or small molecules, and none of these were able to prevent adhesion formation in mice. Thus, we believe that the adhesion molecules that are involved in epithelial cell-cell contacts and which are present normally in healthy mesothelium, do not function or participate in surgical adhesion disease progression. Instead, we propose that membrane protrusions and attachments between cell membranes is the linchpin of the adhesiogenic cascade, which is molecularly different from epithelial cell-cell adhesions. We agree with the reviewer regarding the over-interpretation of fusions, and have omitted this term from the manuscript where not appropriately used.

Figure 3.

7) Fig. S1 shows that stress and clustering of the carrier aggregates is accompanied by a very impressive change in the expression of the adhesion proteins such as CD44 in Met-5A cells. The analysis is carried out 2 days after the stress and the change in the expression is discussed as “bead clustering induces biomarker expression”. It would be interesting to test whether the change in the adhesion protein expression precedes or follows the bead clustering.

Author’s response

Based on the reviewer’s comment, we have now looked into the role of CD44 in more detail, but found inconsistencies between our in vitro and in vivo model. For our bead assay, CD44 expression possibly precedes bead clustering, as it starts to be expressed very rapidly after stress (8 hours after stress), and could effectively prevent adhesions when targeted with a blocking antibody against CD44 (see Figure 3A and B below). However, even though we report increased and rapid expression of CD44 in our in vivo model, targeting CD44 with our blocking antibody had no effect in vivo (see Figure 3C below). In this light, we do not consider CD44 as part of the core adhesion program, but can be used as a biomarker to indicate the presence of adhesions or stressed mesothelia.

Figure 3.

8) Is drastic increase in carrier-to-monolayer aggregation for the Met-5A cells subjected to air or talcum desiccation specific for mesothelial cells or can be observed for other cells, say fibroblasts?

Author's response

Based on the reviewer's comment, we have now tested the effects of desiccation on HEK-293 cells cultured on beads, and did not observe any clustering (see Figure 4 below). Furthermore, we have conducted additional experiments in our in vivo model that ruled out the contribution of fibroblasts in adhesion development in vivo. First, we performed lineage tracing of mesothelial cells with the use of a Procr-Cre(ER) and accompanying mTmG reporter line, showing fibroblast cells in advanced adhesions can be traced back to a mesothelial origin. Second, we performed selective ablation of mesothelial cells in a Procr-specific DTA mouse line. When mesothelial cells were genetically ablated after surgery, adhesions completely failed to develop. These new sets of in vivo experiments have now been added to the revised manuscript.

Figure 4.

Minor comments

1) Please number the figures.

Author's response

We have numbered the figures accordingly.

2) Please format the reference: "Jonathan M Tsai, et al (2018). Surgical adhesions in mice are derived from mesothelial cells and can be targeted by antibodies against mesothelial markers. *Sci. Transl. Med.* 10, eaan6735."

Author's response

We apologize and have corrected this mistake.

Reviewer #2 (Remarks to the Author):

The authors provide fundamental advances from current existing knowledge, into contribution of mesothelial cells to adhesive development. The authors provide comprehensive in vitro studies examining histologic imaging in conjunction with molecular biologic analysis of the pathophysiology of adhesions, along with complementary in vitro studies which also define the timeline of adhesion development.

While demonstrating mesothelial cell contribution to adhesion, the authors may be overstating the singular existence of this pathway, and fail to indicate possible implication of fibroblast contribution. The authors do not assess fibroblast contribution in their in vitro models, and have chosen an abrasion in vivo in which “islands” of mesothelial cells may persist. It would be interesting to examine similar endpoints in a peritoneal excision model, to assess potential dual contributions of mesothelial cells and fibroblasts.

Author’s response

We have now conducted additional in vivo experiments in order to rule out the contribution of fibroblasts in adhesion development. First, we performed lineage tracing of mesothelial cells with the use of the mesothelial marker Procr. Using Procr-Cre(ER) and accompanying mTmG reporter line, we show that fibroblastic cells in advanced adhesions can be traced back to a mesothelial origin (see Figure 2G in the revised manuscript). We did not perform an excision model, but decided instead to selectively ablate mesothelial cells in a Procr-Cre(ER)-DTA mouse line, whereby tamoxifen treatment kills mesothelial cells. When mesothelial cells were ablated immediately after surgery, adhesions completely failed to develop (see Figure 2H-I in the revised manuscript), proving that mesothelial cells are critical in the adhesion process, and that fibroblasts or potential other cell populations alone are insufficient to drive this pathological response.

The authors fail to include among their references many prior reports which have also identified the contribution of mesothelial cells to adhesion formation.

Author’s response

We have now included a more complete reference list to the revised manuscript’s bibliography.

Specific minor comments are:

- While not trying to diminish the potentially severe consequences of adhesions, several statements in the introduction are phrased in misleading ways, making the consequences worse than generally accepted. As an example, three-fourths of abdominal operations have adhesions obstruct the small bowel; occurrence is that high, but “obstruction” is much lower. Similarly, a third of patients being readmitted is four years; the SCAR study looked at a 10 year window.

Author’s response

We have changed the text where appropriate.

- The Tsai reference is not provided

Author's response

We apologize and have added this new reference.

Reviewer #3 (Remarks to the Author):

Review on the manuscript by Fischer et al., "Post-surgical adhesions are caused by membrane bridges and fusions between mesothelial surfaces".

Post-surgical adhesions are still an unmet challenge causing intense morbidity and mortality. Recently, Yuval Rinkevich's lab identified the mesenchymal cell lineage which contributes to the formation of post-surgical adhesions. In the here reviewed manuscript, he and his collaborators now uncovered the underlying mechanisms and how to therapeutically address the currently unresolved problem of post-surgical adhesions. The authors developed a novel in vitro assay to visually monitor the early interactions between organ surfaces in real time and in response to distinct pathogenic factors. If pathogenic factors are applied in this in vitro model of mesothelial cells covered microcarriers, adhesion to an unstressed monolayer of mesothelial cells was instantly initiated resulting in strong adhesion of the Met5A cells grown on the carriers and the underlying monolayer. This process follows a sequence of events, in early phases post injury, proliferation of mesothelial cell occurred, thereafter markers of EMT (epithelial-mesenchymal - transition) were detected including WT1, α -SMA, indicative of a contractile cytoskeleton. This is followed by matrix deposition with fibronectin and collagen type III. To validate these in vitro results, the authors developed a valuable murine model closely mimicking adhesion as occurring in humans. Results originating from this preclinical murine model confirm the results of the in vitro model. Most interestingly, RNA seq analysis assessed an unbiased gene expression, and pathway analysis, identified genes regulating calcium homeostasis and cytoskeleton adaptation. were mainly up-regulated. This, most interestingly, could be confirmed in post-surgical adhesions from human patients. Small molecules interfering either with calcium influx or as with cytoskeleton gene expression after adhesion stimulation could very convincingly, avoid adhesion formation, this is a clear breakthrough in clinical science, and as such I appreciate this manuscript very much.

Major comments

1) I would recommend to the authors to describe in more detail why the mesothelial cell line is appropriate to reflect primary cell lines.

Author's response

We believe these are common properties to all mesothelial surfaces that are affected by adhesions. Because of its wide availability, we decided to work with the Met-5a mesothelial cell line. Although we have used other mesothelial cell lines, we observed that the growth dynamics and doubling rates of Met-5a cells most closely matched that of primary mesothelial cells, whereas other cell lines were more tumorigenic in appearance (much faster growth rate). However, we understand the reviewer's concern in that primary mesothelial cells of abdominal origin would have been more appropriate in the context of our study. To address this issue, we isolated primary mesothelial cells from the abdomen (see 'Methods' in the revised manuscript), and exposed them to the same treatment regime as originally described. The results of this can be found in Extended Figure 9B-F. Under these conditions we observed: 1) development of expansive protrusions, 2) rapid bead clustering

or adherence of beads to a monolayer culture, and 3) pharmacologic targetability and prevention of adhesions through the same panel of small molecules as originally described. These events therefore support the universal nature of mesothelial adhesions, and the validity of our cell line.

2) I am convinced of the therapeutical approach the authors have chosen, and thereby provide a new avenue of treating post-surgical adhesions. As a scientist though, given they have the candidates, I would be even more enthusiastic, if they choose a complementary genetic approach to silence candidate genes in mesothelial cells and thereafter assess the fusion and adhesive capacity in vitro.

Author's response

We agree with the reviewer and have conducted additional experiments in our bead assay where we targeted various core adhesion genes through siRNA (see Figure 5E and Extended Figure 9A in the revised manuscript). We now report that, similar to our pharmacologic approach, genetic silencing of core adhesion genes completely prevent adhesions from developing.

Reviewer #4 (Remarks to the Author):

This manuscript describes a follow-up study on the recent paper by authors (Tsai et al., Sci Transl Med. 2018) that examines early cellular and molecular events that produce pathological mesothelial adhesions. Tsai et al. showed that the earliest cellular event in adhesion formation is the fusion between stress-activated mesothelial cells. Current study builds upon Tsai et al. and identified further aspects of mesothelial cell fusion. Intriguingly, using RNA-seq in an in vitro system with human mesothelial cell line, they identify early stress-response genes - cytoskeletal and calcium effectors. By using an available panel of small molecule modifiers of their top target genes, authors show that they can reduce fusogenic activity of cells in vitro with 4 small molecules and that they have anti-adhesion properties in vivo, in the mouse model.

Despite some intriguing, potentially therapeutically viable leads that the study identifies, there are numerous concerns with: (a) mechanistic depth of the study, (b) artificial nature of the in vitro system and (c) rigor.

1) All of the major findings in this manuscript are derived from an in vitro system using Met-5A human mesothelial cell line. Met-5A cell line is derived in late 1980s by pRSV-T plasmid-induced transformation of mesothelial cells isolated from pleural fluids of non-cancerous individuals. This is a very significant concern. Other than been immortalized, the cell line is specific to pleural cavity, rather than peritoneal cavity. While authors make many extrapolations of their in vitro data on Met-5A cells to peritoneal mesothelium in vivo, it remains unclear just how similar or different pleural and peritoneal mesothelial cells are (even without immortalization).

It is more than likely that many behaviors that Met-5A cells display in vitro are not reflective of the normal and pathological behaviors by peritoneal mesothelial cells in vivo (even though some of the molecular events are suggested to be conserved).

Further, in an in vitro system, Met-5A cells are cultures out of the normal context of peritoneum and all other cells type (immune, fibroblasts) that likely participate and strongly modulate early-stage response by mesothelium to injury are not present in the in vitro system.

Author's response

We believe the adhesion mechanisms we discover here are common to all mesothelial surfaces where adhesions develop. However, we understand the reviewer's concern that primary mesothelial cells of abdominal origin would have been more appropriate in the context of our study. To address this issue, we isolated primary mesothelial cells from the abdomen (see 'Methods' in the revised manuscript), and exposed them to the same treatment regime as originally described. These new results can be found in Extended Figure 9B-F. Under these conditions we observed: 1) development of expansive protrusions, 2) rapid bead clustering or adherence of beads to a monolayer culture, and 3) pharmacologic targetability and prevention of adhesions through the same panel of small molecules as originally described. These events therefore support the universal nature of mesothelial adhesions, and the validity of our cell line.

2) In an in vitro model, authors fairly conclusively showed that Met-5A cells undergo true cell fusion with cytoplasmic exchange early after stress. However, the molecular mechanism of fusion is not truly studied. The early response genes (cytoskeletal and calcium effector) that authors identify via single-cell RNA-seq on Met-5A cells and immunostaining, can be important for activating cellular protrusion-making activity by Met-5As, rather than cell fusion process by the protrusions per se. Further, authors document multiple morphological types of cellular protrusions that stressed Met-5As form, but they did not study if certain (or all) protrusion types initiate cell fusion and how.

Generally, despite identifying some certainly interesting fusion-promoting gene targets, the molecular mechanism of the fusion process is not disclosed.

Author's response

We agree with the reviewer that a thorough investigation on the molecular principles underlying more specific membrane protrusion types in mesothelial cells would be important and relevant. We have already shown that targeting the core adhesion genes can prevent the formation of protrusions (see new Figure 5D in the revised manuscript) and subsequent fusion, and therefore believe these genes act globally to restrict all protrusion types, and subsequent fusion itself. We have now also strengthened our mechanistic findings by treating mesothelial-covered beads with the specific protrusion inhibitor Ciliobrevin, which also resulted in the complete prevention of adhesions (see Figure 5 below). This further argues for membrane protrusions as the core adhesion agent, with cell fusions acting downstream of the initial protrusion event. Although fusions are undeniably a critical step in the adhesion process, our efforts in this manuscript have been mostly directed at protrusions in the more general sense. Unraveling the complete repertoire of fusion genes will require a substantial amount of work that we think would justify a separate publication. We hope we have addressed the reviewer's query.

Figure 5.

3) Authors make strong claims regarding anti-adhesion potential of Bepridil (calcium channel blocker), Rhosin (Rho-GEF inhibitor), CK-666 (Arp2/3 complex and actin

assembly inhibitor), and Golgicide A (Arf1-mediated actin organization inhibitor). These claims are based on in vitro data on Met-5A cells (Fig 4a-c) and on limited in vivo studies in mice (Fig 4d). This is certainly a superficial and preliminary assessment and in order to truly establish any given gene as a critical target in mesothelial fusogenesis, a more comprehensive studies, involving genetic mouse model(s) and gene perturbation studies in vitro in primary peritoneal mesothelial cells (both loss- and gain-of-function) will be necessary. At present, small molecule studies on Bepridil, Rhosin, CK-666 and Golgicide A are fairly preliminary, despite being intriguing.

Author's response

Based on the reviewer's comment, we have now conducted additional mechanistic experiments using our bead assay where we targeted specific core adhesion genes through siRNA (see Figure 5E and Extended Figure 9A in the revised manuscript). We report that, similar to our pharmacologic approach, genetic silencing of core adhesion genes completely prevented adhesions from developing. We have now also conducted additional experiments in our in vivo murine adhesion model. Here, we performed lineage tracing of mesothelial cells with the use of a Procr-Cre(ER) and accompanying mTmG reporter line, showing cells with expansive protrusions in advanced stage adhesions can be traced back to a mesothelial origin (see Figure 2G in the revised manuscript). We also selectively ablated mesothelial cells in a Procr-Cre(ER)-DTA mouse line, whereby tamoxifen treatment kills mesothelial cells. When mesothelial cells were ablated immediately after surgery, adhesions completely failed to develop (see Figure 2H-I in the revised manuscript), indicating that mesothelial cells are the primary agents in the adhesion process, and that fibroblasts or potentially other cell populations alone are insufficient to drive this pathological response. We have additionally checked for the availability of existing transgenic lines in order to knock-out core adhesion genes, but were not successful. We could design viral vectors to target these genes, but this would require a substantial amount of work that will not add to the clinical relevance of the identified therapeutic targets.

4) Study rigor. Although likely addressable, at present manuscript lacks replicate "n" values for all experiments throughout. Statistical analysis is rudimentary (p values are shown on figures, but underlying data is not well defined).

Author's response

We have now clarified the text and figure legends to be more transparent with statistical data. Further, all experiments that have been added now also include appropriate quantifications. Figures that show the mesothelial transition following stress, involve a very strong phenotype that is immediately apparent. We believe that including quantification for these specific figures would therefore add little information in what is available. All new replicates have been added and are indicated as dots in each individual bar chart.

Reviewers' Comments:

Reviewer #1:

Remarks to the Author:

The revision and, especially, the new data showing the complete lack of the adhesions in the absence of mesothelial cells, has considerably strengthened the paper. The results are exciting and unexpected (for instance, the adhesion propagation phenomenon). The paper reads very well, and the findings have important biomedical implications. I have now only a few very minor suggestions.

Specific comments.

1) I still suggest rewriting the sentence "To expand on these findings, we mixed stress-exposed carrier-bound cells with unstressed carrier-bound cells, and labeled them with red or green membrane lipid dyes respectively." to reflect that the stressed carrier-bound cells and unstressed carrier-bound cells were first labeled and then mixed. It can be something like: "To expand on these findings, we labeled carrier-bound cells with red membrane lipid dye and mixed them with unstressed carrier-bound cells labeled with green membrane probe."

2) In the legend to Fig. 1 I see no description of panel f.

3) Line 471 A typo in "following themanufacturer's" should be corrected to "following the manufacturer's".

Reviewer #2:

None

Reviewer #3:

Remarks to the Author:

The authors undertook impressive efforts to address the questions whether the initially employed cell line fully reflects the properties of primary abdominal mesothelial cells. In addition, they went into the mechanisms of initial adhesion events and identified candidates of their RNAseq screen to be causally involved. This is now a very important manuscript with major relevance for clinical medicine.

Reviewer #4:

Remarks to the Author:

1) Authors' revision only partially addressed original major concerns and key issues remain unresolved. Bulk of in vitro work remains on non-peritoneal Met-5A human mesothelial cell line. Authors show some pilot data on primary human peritoneal mesothelial cells (Extended Data Fig 9), but it is preliminary and key mechanistic aspects of the study remain not validated on normal abdominal mesothelium. It is a strong conjecture that the preliminary data in Extended Data Fig 9 is sufficient to definitely state that developmentally and anatomically distinct peritoneal mesothelial cells behave just like pleura-derived Met-5A cell line.

2) No effort was made to address the impact of other cells type (immune, fibroblasts) in the described process. As I stated, other key cell types that surround mesothelium (immune, fibroblasts) likely participate and strongly modulate early-stage response by mesothelium to injury. They are not present in the in vitro system and extrapolating results from one-cell-type in vitro system to a much more complex in vivo system and ignoring the role of innate and adaptive immune cells is incorrect.

2) It is disappointing, but in this revision authors did not meaningfully address how protrusions

result in stable cell fusion. The mechanism of fusion and how (and which) cellular protrusions initiate and mediate fusion process remains unexplored. At this point, authors can claim that membrane protrusions are necessary and temporally proximal to cell fusion, but beyond that the mechanism of what really matters -- successful and stable cell fusion itself -- is unclear. In the response letter, authors show the results of Met-5A cell treatment with Ciliobrevin, which authors define as specific protrusion inhibitor (Figure 5 in the Response). Ciliobrevin indeed significantly inhibits nanoluciferase emission (and by association fusion). Yet, ciliobrevins are a group of small molecule inhibitors of dynein 1 and 2. Their effects on cells are quite diverse and non-specific and a number of intracellular processes that require dynein function, including organelle transport, will be disrupted. Further, ciliobrevins are potent Hedgehog pathway antagonist as they disrupt primary cilia formation. The mechanism of this new experiment is not clear and interpretation is not definitive.

3) New functional results with ProcrCreERT2 mice are intriguing. Authors claim that ProcrCreERT2 is very specific to mesothelium, but reference unpublished data, without showing comprehensive Cre validation. However: a) Based on Rosa-mTmG reporter (Extended Data Figure 5d) there appears to be a lot of non-mesothelial GFP expression both within ceacal epithelium and underneath (possibly stroma, immune or vascular cells?). Thus, the mesothelium specificity of this Cre line is questionable without additional substantial further confirmation. b) Based on Rosa-mTmG reporter, authors observe approximately 50% mesothelial labeling in mice after 3 consecutive Tamoxifen treatments. This is not very high and the expression shown in Extended Data Figure 5c is rather patchy. Yet, after DTA induction with just one tamoxifen dose, very surprisingly authors observed very low "adhesion score" - essentially full rescue of adhesion. This dramatic functional result is very difficult to reconcile with the patchiness of mesothelial ProcrCreERT2 activity even after 3 doses of tamoxifen. It is reasonable to assume that the killing efficiency of mesothelium in the DTA experiment is well below 50%, thus there should be still many mesothelial cells available for fusion. This experiment is not definitive or clean and alternative explanations have not been considered.

Reviewer's comments:

Reviewer #1 (Remarks to the Author):

The revision and, especially, the new data showing the complete lack of the adhesions in the absence of mesothelial cells, has considerably strengthened the paper. The results are exciting and unexpected (for instance, the adhesion propagation phenomenon). The paper reads very well, and the findings have important biomedical implications. I have now only a few very minor suggestions.

Specific comments.

1) I still suggest rewriting the sentence "To expand on these findings, we mixed stress-exposed carrier-bound cells with unstressed carrier-bound cells, and labeled them with red or green membrane lipid dyes respectively." to reflect that the stressed carrier-bound cells and unstressed carrier-bound cells were first labeled and then mixed. It can be something like: "To expand on these findings, we labeled carrier-bound cells with red membrane lipid dye and mixed them with unstressed carrier-bound cells labeled with green membrane probe."

Author's response

We agree and have changed the sentence.

2) In the legend to Fig. 1 I see no description of panel f.

Author's response

The description was there, but incorrectly labelled. It has now been corrected.

3) Line 471 A typo in "following themanufacturer's" should be corrected to "following the manufacturer's".

Author's response

We have corrected the mistake.

Reviewer #3 (Remarks to the Author):

The authors undertook impressive efforts to address the questions whether the initially employed cell line fully reflects the properties of primary abdominal mesothelial cells. In addition, they went into the mechanisms of initial adhesion events and identified candidates of their RNAseq screen to be causally involved. This is now a very important manuscript with major relevance for clinical medicine.

Reviewer #4 (Remarks to the Author):

1) Authors' revision only partially addressed original major concerns and key issues remain unresolved. Bulk of in vitro work remains on non-peritoneal Met-5A human mesothelial cell line. Authors show some pilot data on primary human peritoneal mesothelial cells (Extended Data Fig 9), but it is preliminary and key mechanistic aspects of the study remain not validated on normal abdominal mesothelium. It is a strong conjecture that the preliminary data in Extended Data Fig 9 is sufficient to definitely state that developmentally and anatomically distinct peritoneal mesothelial cells behave just like pleura-derived Met-5A cell line.

Author's response

Our additional experiments that include primary mesothelial cells of the abdomen completely mimic our in vitro pleural findings, and hence strongly hint towards a universal role of the mesothelium with regards to its adhesion properties. To definitively confirm this would require a large battery of experiments from every possible organ surface. It is not in our interest to conduct such a large scale study, nor would it add much value to the scope of this manuscript. We have already phrased our claims more conservatively to more accurately reflect our provided evidence, and we hope that this sufficiently addresses the reviewers concerns.

1) No effort was made to address the impact of other cells type (immune, fibroblasts) in the described process. As I stated, other key cell types that surround mesothelium (immune, fibroblasts) likely participate and strongly modulate early-stage response by mesothelium to injury. They are not present in the in vitro system and extrapolating results from one-cell-type in vitro system to a much more complex in vivo system and ignoring the role of innate and adaptive immune cells is incorrect.

Author's response

Our in vitro assays, in vivo lineage tracing, live imaging of injured mesothelium, and selective ablation of PROCR⁺ mesothelium, collectively, confirms the mesothelium as the critical and major cell player in generating abdominal adhesions, as well as provide a conceptual framework as to how these adhesions develop. Although we agree that other cell types are involved in the adhesion process, we propose that these populations serve mainly to modulate and likely exacerbate the adhesion response. However, it was not within the scope of this manuscript to tease out the individual contribution of every cell type to the adhesion process. We also do not believe this is strictly necessary, as our data clearly indicates an all-encompassing role for the mesothelium. To provide more support for our claims, we have carried out additional immuno-stainings for CD45 (pan-immune marker) and PDGFR α (pan-fibroblast marker) in conjunction with our PROCR-CreER;Rosa26mtmg transgenic mice. Figure 1 outlined below (also included in our revised manuscript, see Extended Figure 5) depicts the area in the center of the adhesion (adhesion core), sampled 5 days after injury when adhesions have fully developed. In this adhesion core, many PROCR⁺ descendants can be appreciated with fibroblast-like morphologies, as we originally describe in the manuscript. A co-staining for CD45 and PDGFR α additionally reveals the presence of immune cells and fibroblasts in the adhesion core, supporting the contribution of these cell types to adhesions as the reviewer states. However, please note that many of the GFP-labelled cells are also PDGFR α positive (whilst being PDGFR α negative under healthy control conditions), indicating that they have adopted fibroblast properties. Thus, while the fibroblast is unmistakably an endpoint player in the adhesion pathology, we believe that the bulk, if not all of these cells have a mesothelial origin. Secondly, none of the GFP-labelled mesothelial cells show marker expression normally attributed to immune lineages, indicated by their lack of CD45 expression, suggesting adhesiogenic cells have no immunological origin.

Figure 1. Scale bar, 50 μ m.

2) *It is disappointing, but in this revision authors did not meaningfully address how protrusions result in stable cell fusion. The mechanism of fusion and how (and which) cellular protrusions initiate and mediate fusion process remains unexplored. At this point, authors can claim that membrane protrusions are necessary and temporally proximal to cell fusion, but beyond that the mechanism of what really matters -- successful and stable cell fusion itself -- is unclear. In the response letter, authors show the results of Met-5A cell treatment with Ciliobrevin, which authors define as specific protrusion inhibitor (Figure 5 in the Response). Ciliobrevin indeed significantly inhibits nanoluciferase emission (and by association fusion). Yet, ciliobrevins are a group of small molecule inhibitors of dynein 1 and 2. Their effects on cells are quite diverse and non-specific and a number of intracellular processes that require dynein function, including organelle transport, will be disrupted. Further, ciliobrevins are potent Hedgehog pathway antagonist as they disrupt primary cilia formation. The mechanism of this new experiment is not clear and interpretation is not definitive.*

Author's response

Although a complete understanding of the fusion mechanism would be interesting and relevant for the topic, we deliberately decided to refrain from diving too deep into the fusion mechanism. Cell fusion is a complex process that requires an elaborate sequence of steps. Understanding the complete fusion response would ask for a host of new experiments,

including, but not limited to: the initial priming response that allows cells to be susceptible for fusion, 2) establishment of cell-cell contact allowing apposing lipid bilayers to attach, 3) initiation of the actual merging of cell membranes, 4) maintenance of stable fusions or uncoupling of fused protrusions allowing detachment, and 5) orchestration of molecular routes that allows the transport of some, but not other proteins into the receiving cell.

Although we can obtain potential fusion candidates from our RNAseq analysis, we believe that in order to facilitate fusion in such a rapid manner, many of these events are driven by the compartmentalization of existing proteins, rather than through de novo gene transcription. For example, amongst our top regulated genes we can find AKAP12 and EZRIN, two proteins who's main function is to bring existing proteins from one location in the cell to another. Obtaining a complete map of the mesothelial fusion repertoire would thus require a proteomics approach, in addition to a myriad of additional experiments to address the fusion mechanisms as described above. Combined, this would be more than sufficient for a separate publication, and it is for that reason primarily that we have not undertaken these steps.

3) New functional results with ProcrCreERT2 mice are intriguing. Authors claim that ProcrCreERT2 is very specific to mesothelium, but reference unpublished data, without showing comprehensive Cre validation. However: a) Based on Rosa-mTmG reporter (Extended Data Figure 5d) there appears to be alot of non-mesothelial GFP expression both within ceacal epithelium and underneath (possibly stroma, immune or vascular cells?).

Author's response

This is incorrect. What you can see in Extended Figure 5d in the cecum are remnants of faeces that have not been washed out during the staining procedure. Faeces is highly autofluorescent in the 488 nm range, accounting for the unspecific labelling observed in the figure. You can see a better example of this in Figure 2G of the main manuscript. Although it is correct that PROCR is not exclusively expressed in the mesothelium, e.g. PROCR can be found in some vascular and stem cells, we have found our line to be highly specific for the mesothelium with regards to the peritoneum and cecum (we have observed other cell populations in different organs, such as the liver).

Thus, the mesothelium specificity of this Cre line is questionable without additional substantial further confirmation. b) Based on Rosa-mTmG reporter, authors observe approximately 50% mesothelial labeling in mice after 3 consecutive Tamoxiphen treatments. This is not very high and the expression shown in Extended Data Figure 5c is rather patchy. Yet, after DTA induction with just one tamoxiphen dose, very surprisingly authors observed very low "adhesion score" - essentially full rescue of adhesion. This dramatic functional result is very difficult to reconcile with the patchiness of mesothelial ProcrCreERT2 activity even after 3 doses of tamoxiphen. It is reasonable to assume that the killing efficiency of mesothelium in the DTA experiment is well below 50%, thus there should be still many mesothelial cells available for fusion. This experiment is not definitive or clean and alternative explanations have not been considered.

Author's response

We have optimized our tamoxifen regime on the conservative side to ensure a robust labelling of cells. In fact, for many animals we can already observe a near 50% coverage after a single 2 mg tamoxifen dose (see Figure 2 below). However, because Cre expression can be quite variable even among litter-mates, we opted for three injections. Why is there only a 50% coverage after three injections? We cannot definitely answer this, but we are dealing with simple probability here. There will be fewer unlabeled cells after every round of

injection, so the coverage will naturally plateau over time. Additionally, there may be heterogeneity within the mesothelial surface layer that could differ in their sensitivity to Cre, thus limiting the number of recombination events. Furthermore, we cannot rule out that adhesion stimuli activates metabolic pathways in injured mesothelium that could result in a faster uptake of tamoxifen.

Why can we prevent adhesions when we only ablate 50% of mesothelial cells? First, we don't think that one can extrapolate the results of our PROCR reporter line to our PROCR DTA line. The amount of Cre required to sufficiently express eGFP in ways that are visible to the eye may be drastically different compared to the amount necessary to recombine the DTA cassette, causing ablation of the Cre-expressing cell. In other words, a single injection of tamoxifen may ablate much more cells in our DTA line compared to our mTmG line. Regardless, even if only 50% of cells were to be ablated, this is perfectly feasible. It is likely that a specific threshold in terms of number of adhering cells need to be met in order to establish a stable adhesion. Any number of cells below this threshold will not result in adhesions, because of counteracting forces that act against the adhering cells (e.g. peristaltic movements). It is reasonable to assume that this threshold lies anywhere near 50%.

Figure 2. Tissue sampled 48 hours after tamoxifen injection.

Reviewers' Comments:

Reviewer #4:

Remarks to the Author:

In this revision, authors failed to meaningfully address major criticism and my original concern that the key experimental data in the paper is over-interpreted and mis-interpreted is not reduced.

1) In response to the criticism regarding the use of non-peritoneal Met-5A human mesothelial cell line to generate major conclusions on peritoneal fusion mechanism, with only very limited validation in primary human peritoneal mesothelial cells - authors argue that new experiments on primary peritoneal mesothelium cells will be too time-consuming and are unnecessary. This is equivalent to saying, for example, that commonly used HEK293 human embryonic kidney cell line can be used to study the biology of normal kidney tubule cells or podocytes. Any cell line is highly epigenetically and functionally modified compared to their cell source of origin.

2) In response to the request to further determine how stable mesothelial cell fusion occurs - which is, in fact, functionally very critical to the proposed mechanism - authors similarly argue that this will be too time-consuming and, thus, unnecessary for this paper.

3) In response to the potential role of non-mesothelial cells (fibroblasts and immune cells) in peritoneal fusion - authors again argue that this is outside the scope of the manuscript. Yet, new Figure 1 in the response (also Extended Figure 5f) shows presence of large numbers of both immune cells (CD45+) and fibroblastic cells (PDGFRa+) in the peritoneal fusion on day 5 - clearly suggesting their importance.

Further, this statement by authors is inconsistent with data shown:

"However, please note that many of the GFP-labelled cells are also PDGFRa positive ... indicating that they have adopted fibroblast properties. Thus, while the fibroblast is unmistakably an endpoint player in the adhesion pathology, we believe that the bulk, if not all of these cells have a mesothelial origin."

The last part of this statement is inconsistent with the data shown in Extended Figure 5f. I carefully examined this image, cell-by-cell, and only ~1/3 of PDGFRa+ cells have significantly overlapping GFP signal. Therefore, majority of fibroblasts in adhesions are not the result of mesothelial cell EMT.

This issue was already raised in authors' 2018 Science Trans Med paper (DOI: 10.1126/scitranslmed.aan6735) and the Discussion states:

"Therefore, our study cannot exclude the fact that other cells such as submesothelial fibroblasts may also contribute to adhesions. Further studies need to be done to specifically label these cells genetically or chemically and then to trace these cells after adhesion induction surgery to document their contributions to adhesion formation..."

The discussion then goes on to bring up the very same issue brought up in my criticism about the need to determine the role of intestinal stroma fibroblasts and immune cells; yet, authors now insist that this should not be studied in this follow up manuscript.

4) On the reason why adhesion phenotype is completely suppressed in DTA experiments given low efficiency of PROCR-CreER in mesothelium - authors basically state that they do not really know. They also state that the level of induced DTA can be much higher than the level of induced GFP. Yet, this is an impossible explanation, since both the DTA and mTmG constructs are driven by highly ubiquitous ROSA promoter. DTA or GFP activation in this model is an OFF/ON-type of event driven by an irreversible Cre-Lox recombination event. Therefore, the following explanation by authors is improbable: "The amount of Cre required to sufficiently express eGFP in ways that are visible to the eye may be drastically different compared to the amount necessary to recombine

theDTA cassette."

Taken together, in my opinion authors are not prepared to meaningfully address key gaps in their work; they downplay the functional importance of fibroblasts and immune cells in fusion and do not provide meaningful explanation for the key in vivo fusion rescue phenotype of PROCR-CreER;DTA mice.

REVIEWERS' COMMENTS:

Reviewer #4 (Remarks to the Author):

In this revision, authors failed to meaningfully address major criticism and my original concern that the key experimental data in the paper is over-interpreted and mis-interpreted is not reduced.

1) In response to the criticism regarding the use of non-peritoneal Met-5A human mesothelial cell line to generate major conclusions on peritoneal fusion mechanism, with only very limited validation in primary human peritoneal mesothelial cells - authors argue that new experiments on primary peritoneal mesothelium cells will be too time-consuming and are unnecessary. This is equivalent to saying, for example, that commonly used HEK293 human embryonic kidney cell line can be used to study the biology of normal kidney tubule cells or podocytes. Any cell line is highly epigenetically and functionally modified compared to their cell source of origin.

Author's response

The reviewer glosses over the fact that we have performed a number of important experiments, and makes it seem that we have not performed any validation at all. On the contrary, our primary abdominal mesothelium experiments have shown: 1) similar binding kinetics to micro-carriers as compared to our cell line, 2) similar protrusion development, and 3) similar targetability and prevention of binding. These experiments recapitulate all of the major adhesion properties we have documented with the exception of cell fusion, and are therefore the opposite of 'very limited'. Moreover, to make the comparison with HEK293 cells and conclude it as 'equivalent' is far-fetched in our opinion. HEK293 cells have a kidney embryonic origin and currently used cells display no evident tissue-specific gene expression. Their gene signature most closely resembles that of adrenal and neuronal cells (DOI: 10.1016/j.gene.2015.05.065). HEK293 cells furthermore have an unstable karyotype and display variable tumorigenic potential. To compare them with adult podocytes would be absurd. Met-5A cells on the other hand have been derived from adult healthy pleural fluid, are mesothelial in origin and are non-tumorigenic (Ke Y et al. Am J Pathol. 1989). To make the comparison with adult abdominal mesothelium is not unorthodox.

Further, this statement by authors is inconsistent with data shown:

"However, please note that many of the GFP labelled cells are also PDGFR α positive ... indicating that they have adopted fibroblast properties. Thus, while the fibroblast is unmistakably an endpoint player in the adhesion pathology, we believe that the bulk, if not all of these cells have a mesothelial origin."

The last part of this statement is inconsistent with the data shown in Extended Figure 5f. I carefully examined this image, cell-by-cell, and only ~1/3 of PDGFR α + cells have significantly overlapping GFP signal. Therefore, majority of fibroblasts in adhesions are not the result of mesothelial cell EMT.

Author's response

The point here is not that all PDGFR α + cells are also GFP+, the point is that all PDGFR α + cells are GFP+, indicating that all tagged mesothelia undergo EMT in response to adhesion stimuli. The remaining PDGFR α + cells that are GFP- are either derived from mesothelia as well, but were not originally hit with tamoxifen (remember we achieved a 50% labelling coverage), or they are derived from a non-mesothelial source. It is therefore incorrect to state that only 1/3rd of the PDGFR α + cells have a mesothelial origin. Again, we do not refute the claim that there are non-mesothelial cells contributing to adhesions, but the exact contribution of these cells will have to be determined empirically, and in all likelihood is minimal.

This issue was already raised in authors' 2018 Science Trans Med paper (DOI: 10.1126/scitranslmed.aan6735) and the Discussion states:

"Therefore, our study cannot exclude the fact that other cells such as submesothelial fibroblasts may also contribute to adhesions. Further studies need to be done to specifically label these cells genetically or chemically and then to trace these cells after adhesion induction surgery to document their contributions to adhesion formation..."

The discussion then goes on to bring up the very same issue brought up in my criticism about the need to determine the role of intestinal stroma fibroblasts and immune cells; yet, authors now insist that this should not be studied in this follow up manuscript.

Author's response

As the reviewer correctly states in our previous 2018 Science Trans Med paper we documented the role of the mesothelium in adhesions, and addressed the critical aspect of hypoxia in this response. However, we did not touch upon any possible cellular mechanism, which is why we performed this follow up work. In this context, we discovered a unique mechanism that explains the rapid tethering of organs through a dramatic transformation of mesothelia, and the expansive membrane network that accompanies it. This was the core focus of this paper, not the contribution of different lineages to the adhesion pathology, which we only added upon request by the reviewer's. Although we believe these experiments serve to solidify our findings and improve the paper overall, we do not think that a more extensive, exhaustive interrogation on the different lineage contributions in the adhesion pathology fits the scope of this paper, as this paper was never meant to pursue that goal.

3) In response to the potential role of non-mesothelial cells (fibroblasts and immune cells) in peritoneal fusion - authors again argue that this is outside the scope of the manuscript. Yet, new Figure 1 in the response (also Extended Figure 5f) shows presence of large numbers of both immune cells (CD45+) and fibroblastic cells (PDGFRa+) in the peritoneal fusion on day 5 - clearly suggesting their importance.

Author's response

Non-mesothelial fibroblasts and immune cells undoubtedly serve an important role in the adhesion cascade, but the mere presence of these cells in adhesion tissue does not establish causality. As we have stated before, we believe these cells have an important modulatory role in the adhesion process, but do not initiate the pathology.

4) On the reason why adhesion phenotype is completely suppressed in DTA experiments given low efficiency of PROCR-CreER in mesothelium - authors basically state that they do not really know. They also state that the level of induced DTA can be much higher than the level of induced GFP. Yet, this is an impossible explanation, since both the DTA and mTmG constructs are driven by highly ubiquitous ROSA promoter. DTA or GFP activation in this model is an OFF/ON-type of event driven by an irreversible Cre-Lox recombination event. Therefore, the following explanation by authors is improbable: "The amount of Cre required to sufficiently express eGFP in ways that are visible to the eye may be drastically different compared to the amount necessary to recombine the DTA cassette."

Author's response

We may have not been clear with our previous explanation. Although the reviewer is correct in that DTA or GFP activation is an OFF/ON-type event, the efficiency of Cre recombination depends on the location of the LoxP sites in the genome and the distance separating them (DOI: 10.1093/embo-reports/kve064). Thus, one should expect a different recombination efficiency for every floxed allele, in our case DTA or mTmG, even when using the same CreERT2-expressing mouse line.

2) In response to the request to further determine how stable mesothelial cell fusion occurs - which is, in fact, functionally very critical to the proposed mechanism - authors similarly argue that this will be too time-consuming and, thus, unnecessary for this paper.

Taken together, in my opinion authors are not prepared to meaningfully address key gaps in their work; they downplay the functional importance of fibroblasts and immune cells in fusion and do not provide meaningful explanation for the key in vivo fusion rescue phenotype of PROCR-CreER;DTA mice.

Author's response

We are sorry to hear that we cannot satisfy the reviewer in spite of a considerable number of experiments that we have carried out. Nonetheless, we have addressed these concerns previously, and feel that at this point, reluctantly, perhaps we should agree to disagree.